# DSGYM: A Holistic Framework for Evaluating and Training Data Science Agents

**Fan Nie** [* 1 2]  **Junlin Wang** [* 2 3]  **Harper Hua** [* 1]  **Federico Bianchi** [2]  **Yongchan Kwon** [2]  **Zhenting Qi** [4]
**Owen Queen** [1]  **Shang Zhu** [2]  **James Zou** [1 2]

## Abstract

Data science agents promise to accelerate discovery and insight-generation by turning data into executable analyses and findings. Yet existing data science benchmarks fall short due to fragmented evaluation interfaces that make cross-benchmark comparison difficult, narrow task coverage and a lack of rigorous data grounding. In particular, we show that a substantial portion of tasks in current benchmarks can be solved without using the actual data. To this end, we introduce DSGYM, a standardized framework for evaluating and advancing data science agents in self-contained execution environments. Unlike static benchmarks, DSGYM provides a modular architecture that makes it easy to add tasks, agent scaffolds, and tools, positioning it as a live, extensible testbed. We curate DSGYM-TASKS, a holistic task suite that standardizes and refines existing benchmarks via quality and shortcut solvability filtering. We further expand coverage with (1) DSBIO: expert-derived bioinformatics tasks grounded in literature and (2) DSPREDICT: challenging prediction tasks spanning domains such as computer vision, molecular prediction, and single-cell perturbation. Beyond evaluation, DSGYM enables agent training via an execution-verified data synthesis pipeline. As a case study, we build a 2,000-example training set and trained Qwen3-4B model in DSGYM that outperforms GPT-4o on standardized analysis benchmarks. Overall, DSGYM enables rigorous measurement of whether agents can plan, implement, and validate data analyses in realistic scientific context.

---

[*]Equal contribution  [1]Stanford University [2]Together AI [3]Duke University [4]Harvard University. Correspondence to: Fan Nie <niefan@stanford.edu>, James Zou <jamesz@stanford.edu>.

*Proceedings of the $43^{rd}$ International Conference on Machine Learning*, Seoul, South Korea. PMLR 306, 2026. Copyright 2026 by the author(s).

## 1. Introduction

Data science serves as the computational engine of modern scientific discovery (Wang et al., 2023). From identifying gene markers to predicting molecular properties, data science workflows turn datasets and scientific hypotheses into empirical evidence. This process often requires heavy coding and tedious interactive analysis (Egg et al., 2025), making it a natural target for Large Language Model (LLM) agents (Wang et al., 2024a) that can automate these labor-intensive but structured tasks and accelerate scientific progress (Boiko et al., 2023; Chen et al., 2025; Sun et al., 2025). Yet reliable automation demands a central requirement beyond textual reasoning: an agent's decisions must be grounded in data and validated by execution.

Evaluating LLMs as data science agents remains challenging. The required skill set for data science is inherently broad, spanning iterative exploration, statistical inference, modeling, and domain-specific toolchains. Existing benchmarks can only capture fragments of this space, and they often differ in task formats, scoring conventions and execution environments (Jing et al., 2024; Majumder et al., 2024; Zhang et al., 2025; Huang et al., 2024). These inconsistencies make integration costly and hinder fair reproducible cross-benchmark comparison. More fundamentally, we revisit a core assumption underlying current data science agent evaluation that file-grounded benchmarks (i.e., tasks accompanied by dataset files) necessarily measure data-dependent reasoning. We observe that a substantial portion of tasks in current file-grounded benchmarks can be solved even without accessing the files, revealing prompt-only shortcuts that inflate performance and confound measurement. Such shortcuts can arise from strong priors, pattern matching, or inadvertent contamination, undermining the validity of file-grounded evaluation as a proxy for genuine data interaction. Moreover, current evaluations under-represent domain-specific scientific workflows, limiting our understanding of agents' ability to support real scientific discovery.

To provide better support for the community, we propose DSGYM, an integrated framework that unifies diverse data science evaluation suites behind a single API. We abstract the complexity of code execution behind containers that can

be allocated in real time to execute code in a safe manner, allowing users to effectively run evaluations even on their local setups. Beyond providing a common execution layer, DSGYM adopts a modular design that makes it straightforward to add new tasks, agent scaffolds, tools and evaluation scripts. This positions DSGYM as a live, continuously extensible testbed for the community to measure and develop data science agents.

Beyond infrastructure, DSGYM contributes DSGYM-TASKS, a curated and expanded task ecosystem. We unify and audit representative benchmarks under a standardized schema, and introduce a *shortcut filtering* to remove tasks that can frequently be solved without data access. This yields a suite where performance more faithfully reflects data-dependent reasoning rather than prompt-only shortcuts. We further expand the evaluation scope by two novel task suites: (i) DSBIO: an expert-derived scientific analysis suite of 90 bioinformatics tasks grounded in academic literature, probing domain-specific scientific reasoning and tool use, and (ii) DSPREDICT: end-to-end modeling challenges sourced from recent Kaggle competitions spanning computer vision, molecular prediction, single-cell perturbation and so on, evaluating whether agents can build functional pipelines and iteratively improve predictive performance.

Using DSGYM, we benchmark frontier proprietary and open-weight LLMs across general analysis, scientific workflows, and end-to-end modeling. We find that even frontier models substantially underperform on scientific workflows: over 80% of annotated failures are due to domain-grounding errors, such as misinterpreting domain concepts or using domain-specific libraries incorrectly. We further identify two recurring model behaviors, *simplicity bias* and lack of verification, that become especially damaging for realistic modeling tasks: on the hard split of DSPREDICT, the *medal* rate is near 0% even though the *valid submission* rate exceeds 60%, suggesting models frequently stop at runnable but under-optimized solutions. Finally, although DSGYM is primarily an evaluation framework, it can also support agent training. We demonstrate this by reusing DSGYM's agent and execution environments to generate execution-verified synthetic queries and trajectories, enabling a 4B model to reach competitive performance with frontier models such as GPT-4o on standardized analysis benchmarks. In summary, **our contributions** are as follows:

- We show that existing data science benchmarks are vulnerable to shortcuts where agents can solve the task without using the actual data.
- We introduce DSGYM, a unified, reproducible framework with standardized abstractions that enables cross-benchmark execution behind a single API.
- We release DSGYM-TASKS, a curated task ecosystem that standardizes and audits representative benchmarks,

filters shortcut-solvable tasks, and expands coverage with DSBIO and DSPREDICT.
- We benchmark frontier proprietary and open-weight LLMs on DSGYM and analyze strengths and failure modes, revealing persistent gaps in domain-specific scientific workflows and common behaviors such as simplicity bias and insufficient verification.
- We demonstrate that DSGYM enables data synthesis for finetuning and substantially improves a 4B model on analysis benchmarks.

## 2. DSGYM: A Unified Framework for Reproducible Data Science Agents

Existing data science benchmarks evaluate useful but *isolated* skills (e.g., statistical reasoning, basic pandas/numpy usage) under heterogeneous execution setups, hindering holistic assessment and reproducible cross-benchmark comparison. DSGYM unifies evaluation in a containerized framework that standardizes tasks, agent interfaces, and execution under a single protocol. Our design is guided by three core requirements:

**(1) Realistic, data-dependent execution.** Tasks require programmatic access to real data files, which necessitates isolated execution, persistent state, resource limits, and strict filesystem separation to prevent shortcuts.

**(2) Cross-benchmark standardization.** To enable fair comparison across tasks from diverse domains, DSGYM standardizes task prompts, answer formats, evaluation metrics, and environment definitions, removing inconsistencies arising from heterogeneous original benchmarks.

**(3) Modularity and extensibility.** A modern testbed must support continuous growth rather than being a fixed static dataset. DSGYM's modular design makes it easy to add new tasks, agent scaffolds, and metrics. The same infrastructure also supports synthetic data generation, enabling research on training data science agents (Section A).

We model data science as a standardized interaction loop where an *Agent* solves a *Task* by interacting with the *Environment* (Fig. 2). We describe the three components below.

### 2.1. Tasks and Datasets

**Task Taxonomy.** DSGYM focuses on the *data-driven investigation* phase of scientific discovery, where hypotheses are tested through analysis and modeling over empirical data. We consider two task types: *Data Prediction*, which learns from $D_{\text{train}}$ and predicts on $D_{\text{test}}$ evaluated by a metric $m$ (e.g., RMSE); and *Data Analysis*, which answers queries via programmatic analysis over one or more datasets $D = \{D_i\}$ (e.g., statistical testing, causal inference, regression).

**Unified Task Abstraction.** Regardless of category, DS-

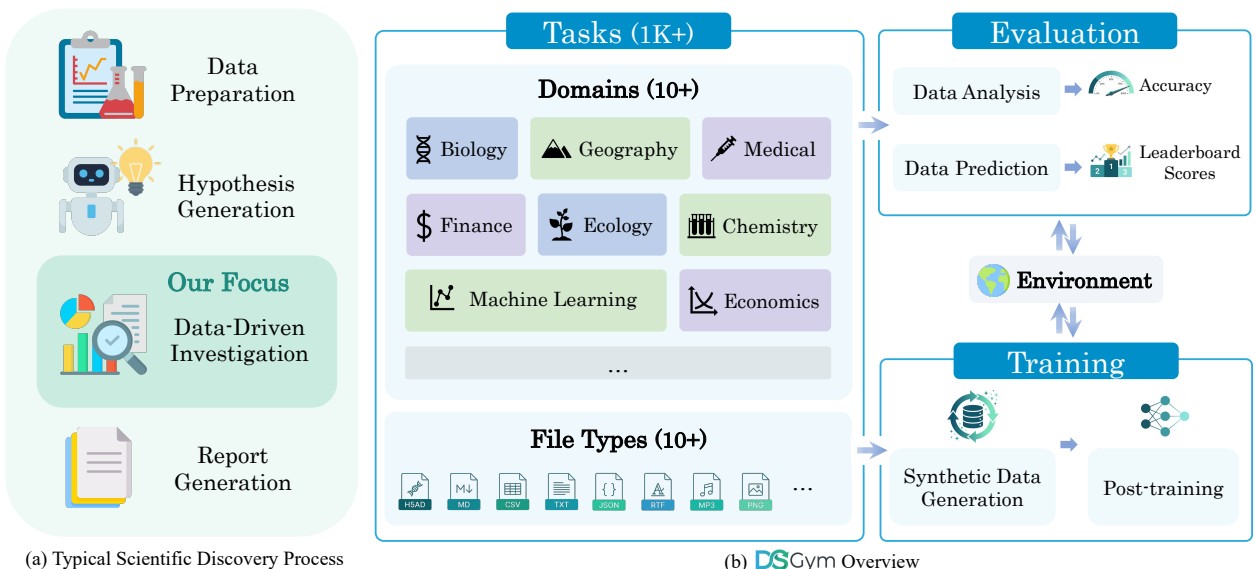

(a) Typical Scientific Discovery Process    (b) DSGym Overview

*Figure 1.* (a) In the typical scientific discovery process, DSGYM specifically focuses on the Data-Driven Investigation phase, where agents must bridge scientific hypotheses and empirical evidence through complex analysis. (b) We provide a unified environment spanning 10+ scientific domains and diverse file types. The framework enables a closed-loop ecosystem for both evaluation and training.

GYM represents each task as a standardized *Task Object* $(\mathcal{D}, \mathcal{P}, \mathcal{M}, \mathcal{Z})$, where $\mathcal{D}$ are required data files (e.g., `.csv`, `.h5ad`), $\mathcal{P}$ is the query prompt, $\mathcal{M}$ defines the evaluation metric, and $\mathcal{Z}$ contains structured metadata such as task category, domain labels, and keyword tags.

**Dataset Organization.** A *Dataset* groups tasks with shared domains and evaluation protocols, simplifying the integration of new tasks and benchmarks.

### 2.2. Agents

The *Agent* wraps a base LLM and maps the history of actions and observations to the next reasoning and action. While DSGYM allows custom agent architectures, we provide a default interface (Fig. 2(b)) In each step, the agent outputs decision blocks in specific tags:

- `<reasoning></reasoning>` for articulating analytical plans or reflecting on progress,
- `` for writing executable Python code to perform analysis or call tools.
- `<answer></answer>` for submitting the final solution when the analysis is complete.

Executable outputs are returned in `<information>` tags. This standardized interface ensures that models are evaluated on their reasoning and coding capabilities independent of agent architecture and prompt design choices.

### 2.3. Environment

A data science framework should offer reproducible environments, controllable resources allocations and the execution traces. In particular, data science workflows de-

mand persistent memory state to efficiently manipulate large datasets. DSGYM runs each agent trajectory inside a dedicated Jupyter kernel hosted within an isolated container. We adopt Jupyter due to its wide adoption in data science, but such a framework could be easily extended to RStudio or other environments. The execution system follows a *manager–worker* architecture for executing actions on separated environments. A central manager container orchestrates the entire system by allocating a fresh worker container for each trajectory, binding read-only dataset mounts and writable workspace, and routing code requests. Each worker hosts an independent Jupyter kernel, ensuring complete isolation. Our execution system has the following features:

- **Stateful execution.** The environment preserves state across interaction steps: variables, trained models, and intermediate files generated in previous turns remain accessible in subsequent ones unless explicitly cleared. Resource limits on CPU, memory, and wall-clock time are enforced per container and can be user-specified. The answers or generated artifacts are extracted and evaluated against the metric in an independent process.
- **Tool integration.** The environment supports *code-represented tools* that are functions callable from the agent's Python code and executed in-kernel. The current release includes a web-search tool as an example, and users can register new tools without altering the system.
- **Domain-specific containers.** Workers may use different container images to satisfy domain-specific dependencies and tools; the manager assigns each task to the appropriate container type, enabling heterogeneous tasks within a

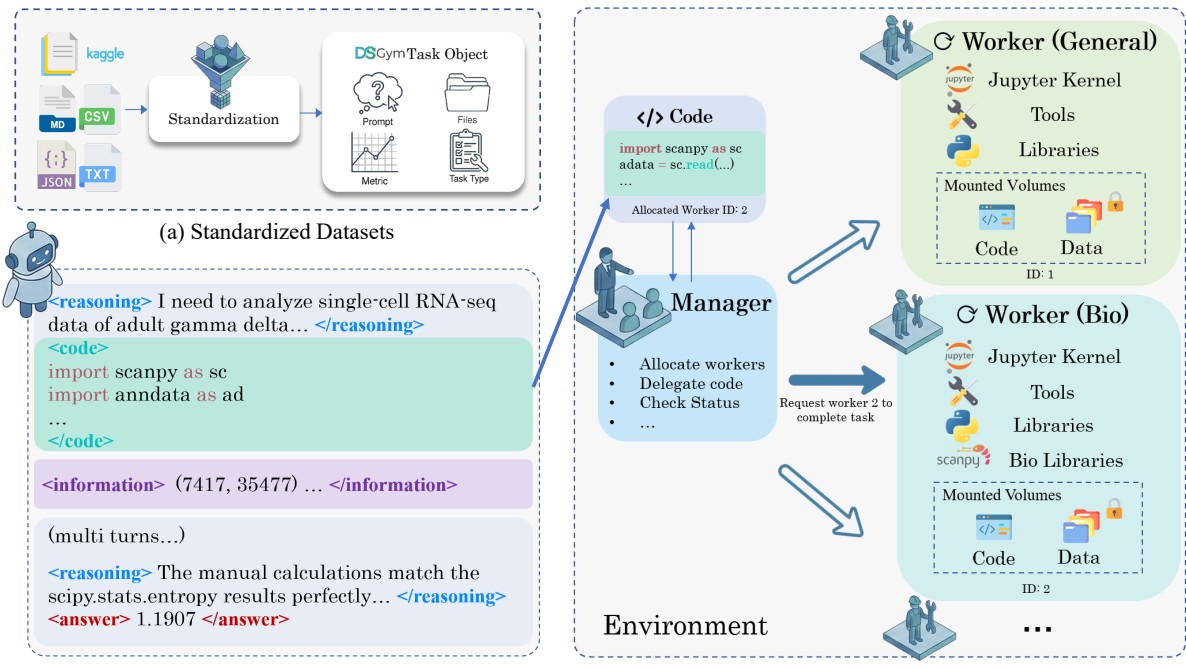

*Figure 2.* **The Architecture of DSGYM.** (a) **Standardized Tasks:** We aggregate heterogeneous data sources into a unified task object. (b) **Agent Interface:** DSGYM provides a default CodeAct-like agent to interact with the environment. (c) **Execution Environment:** A central Manager container orchestrates the execution, dispatching agents to isolated Docker containers (Workers) pre-loaded with domain-specific libraries. Crucially, datasets are mounted as *Read-Only Volumes*, while agents operate in a separate writable workspace.

unified infrastructure.

- **Filesystem Protection.** The environment enforces strict filesystem permissions. Dataset files are mounted to the container's volumes with **read-only** permissions. Agents operate in a separate, isolated writable workspace.
- **Environment Cycling.** Environments can be restarted and cycled, allowing users to choose parallelism and define batching mechanisms.

This architecture enables DSGYM to execute hundreds of trajectories in parallel while maintaining strict isolation, providing a scalable foundation for both evaluation in parallel and training of data science agents.

## 3. Limitations of Existing Data Science Benchmarks

Evaluating LLMs as data science agents requires moving beyond simple code generation to measuring execution-grounded reasoning and analysis pipelines over real-world datasets. While pioneering, existing benchmarks often fail to fully capture this process. Our audit of current benchmarks reveals three systemic limitations that hinder rigorous evaluation:

**Lack of Rigorous Data Grounding.** A core assumption of current file-grounded benchmarks is that tasks requir-

ing data files necessarily measure data-dependent reasoning. However, our analysis reveals a pervasive "shortcut" phenomenon: *many questions can be answered correctly without reading the data.* As shown in Figure 3, across three prominent benchmarks, agents consistently achieve substantial accuracy even when data files are withheld. This suggests that performance is often inflated by data contamination, superficial pattern matching or domain priors rather than genuine interaction with the data.

**Task Invalidity and Inconsistency.** Several widely adopted benchmarks contain issues such as annotation errors, mismatched question–answer pairs, vague formatting instructions, or ambiguous multiple-choice options.

**Limited Operation and Domain Coverage.** As illustrated in Figure S3, existing suites overrepresent general statistics (e.g., descriptive statistics, aggregations, small models) while underrepresenting domain-specific scientific workflows, including specialized terminology, scientific modalities (e.g., `.h5ad`), and domain libraries.

## 4. DSGYM-Tasks

While Section 2 describes the unified architecture, we now turn to the *dataset layer* of DSGYM. DSGYM-TASKS is designed to address the limitations identified in Section 3

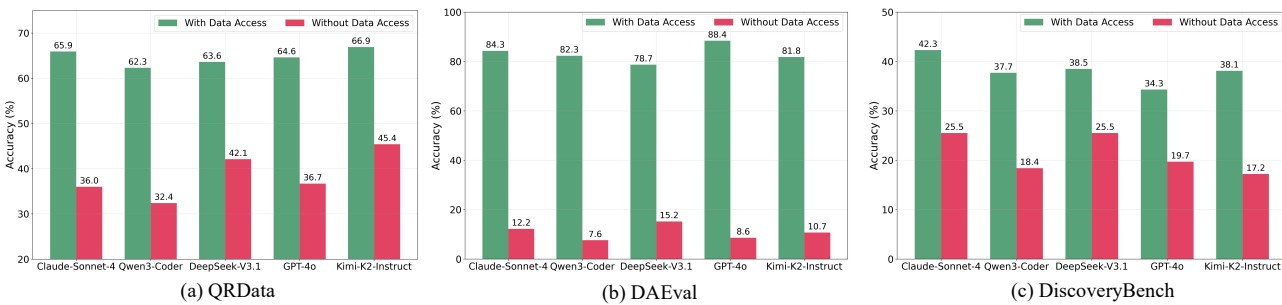

(a) QRData    (b) DAEval    (c) DiscoveryBench

*Figure 3.* **Accuracy with or without data access on three file-grounded benchmarks.** We observe that even when real data files are not provided, agents can still answer a substantial fraction of questions correctly, suggesting that existing benchmarks can be partially solved via memorization, pattern matching, or priors rather than genuine data interaction.

and spans both *general* data science problems that represent the classic analysis workflows familiar to practitioners, and *domain-specific scientific* tasks. The curation of DSGYM-TASKS are guided by three principles: (i) Addressing flaws and inconsistencies in existing datasets; (ii) Enforcing genuine data interaction; and (iii) Expanding operational and domain diversity. All tasks are executed in the containerized environment using the unified task abstraction (Section 2), ensuring fair, reproducible cross-domain evaluation.

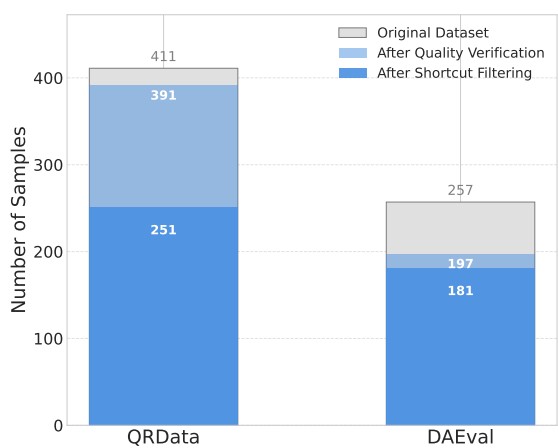

*Figure 4.* Filtering statistics after two-stage refinement.

### 4.1. Refinement of Existing Datasets

We begin by incorporating several widely used benchmarks into DSGYM through two-stage refinement pipeline:

- **Quality verification:** We manually review every item, removing samples that are unscorable, ambiguous, or inconsistent with their gold answers. Formatting issues (e.g., rounding precision, delimiter inconsistencies) are corrected to ensure deterministic evaluation.

- **Shortcut filtering:** To operationalize data-dependence, we run five LLMs from different model families on the remaining tasks *without access to data files* (Figure 3). If a majority ($\geq 3$) of models answer correctly, we mark the task as shortcut-solvable and exclude it from the final

suite. This procedure filters out tasks frequently solvable without interacting with the provided data, including cases driven by memorization, domain priors, or surface-level heuristics, thereby retaining tasks that more directly require execution-grounded reasoning over data files.

Figure 4 summarizes the two-stage refinement. We include four refined datasets: **DAEval-Verified** provides short general-purpose data analysis queries after removing items with missing/misaligned ground truths and enforcing deterministic answer formats. **QRData-Verified** targets statistical and causal reasoning over tabular data after filtering ambiguous multiple-choice questions. **DABStep** contains multi-step financial analysis that requires multi-hop reasoning across multiple data files. **MLEBench-Lite** serves as a canonical data prediction benchmark integrated with our unified environment and metric registry. More details are provided in Appendix E.1.

### 4.2. Scientific Analysis Tasks from Academic Literature

To extend DSGYM beyond generic data analysis, we curate DSBIO, a new suite of **90 bioinformatics tasks** derived from top-tier peer-reviewed publications and open-source scientific datasets. We strategically select Bioinformatics as a *pilot domain* to operationalize scientific discovery, as it uniquely combines high-dimensional, noisy data modalities that demand careful data inspection and domain-grounded statistical reasoning. These tasks probe critical dimensions of competence often underrepresented in existing benchmarks: (1) interpreting unfamiliar data modalities (e.g., gene-expression matrices, spatial omics, high-dimensional noisy data), (2) understanding domain-specific terminology and analytical conventions, and (3) executing workflows with specialized libraries. Each task is constructed through an expert-centered curation process: tasks are derived from published analyses and public datasets, reviewed for biological validity and data-grounded solvability, and paired with executable reference solutions for verification. We intentionally prioritize scientific realism and construction

quality over benchmark scale, making DSBIO an initial high-quality pilot suite rather than a comprehensive coverage of all bioinformatics workflows. Figure S2 demonstrates the data construction pipeline. More details about DSBIO including detailed construction process, expert review, examples are in Appendix E.2.

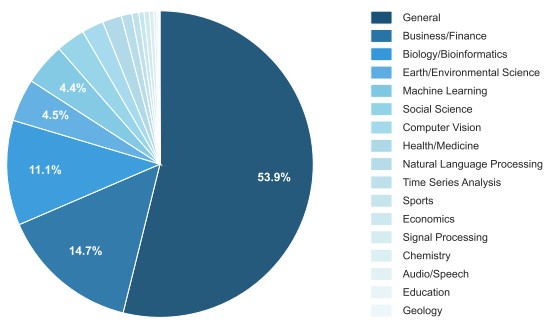

*Figure 5.* Percentage of task domains in DSGYM-TASKS

| Type | Data Analysis | Data Prediction |
|---|---|---|
| Number | 972 | 114 |

*Table 1.* Statistics of DSGYM-TASKS.

### 4.3. Data Prediction Tasks from Kaggle Competition

To capture realistic end-to-end modeling workflows, we implement a fully automated pipeline that continuously collects, filters, and standardizes latest Kaggle competitions into DSGYM. The pipeline consists of three sequential stages: (i) competition crawling, (ii) rule-based filtering, and (iii) dataset preparation ( Figure S2). Details of the construction pipeline are in Appendix E.4. The resulting suite, DSPREDICT, is released in two difficulty splits.

**DSPredict-Easy:** contains 38 simple competitions, primarily from the Kaggle *Playground Series* and two canonical introductory datasets—*Titanic: Machine Learning from Disaster* and *House Prices: Advanced Regression Techniques*. These challenges are intentionally simple in both data structure and task objectives, serving as entry-level testbeds.

**DSPredict-Hard:** contains 54 high-complexity challenges from diverse domains, collecting by our construction pipeline (Section E.4). For challenges with multiple stages, we consistently used the second stage, where the official leaderboard metric is defined. The final dataset suite preserves the original leaderboard metric definitions while ensuring full compatibility.

## 5. Evaluation

We evaluate state-of-the-art LLMs on DSGYM to assess frontier performance across general data analysis, domain-specific scientific workflows, and end-to-end modeling.

### 5.1. Evaluation Setup

**Models.** We evaluate a suite of closed-source models and open-weights models. Unless otherwise specified, all models are evaluated using the default **CodeAct** agent provided in DSGYM with temperature $T = 0$. Although DSGYM environment supports tool integration (e.g., web search), all tools are disabled in all evaluations.

**Metrics.** For analysis tasks, we report exact-match accuracy with a slight numerical tolerance. For prediction tasks, we employ competition-specific leaderboards to derive metrics: Valid Submission Rate, Above Median Rate, Any Medal Rate. For DSPREDICT-EASY, where medal rates are uninformative due to leaderboard saturation, we report Percentile rank instead. More details on metrics and evaluation protocol can be found in Appendix. G.3.

### 5.2. Evaluation Results

Table 2 presents the accuracy on standardized benchmarks. Notably, KIMI-K2-INSTRUCT and CLAUDE 4.5 SONNET perform relatively better than other models. On the expert-derived DSBIO suite (Table 3), performance is consistently lower than on general tasks. Notably, KIMI-K2-INSTRUCT achieves the best overall performance (43.33%), followed by CLAUDE 4.5 SONNET, showcasing their relative robustness in utilizing specialized bioinformatics toolchains. Finally, we assess end-to-end modeling capabilities in Table 4. On MLEBENCH-LITE and DSPREDICT-EASY, most frontier models achieve a near-perfect *Valid Submission Rate* ($> 80\%$), proving that they can construct functional data pipelines. However, on **DSPREDICT-HARD**, even producing a valid submission remains a bottleneck, with most models failing to exceed 70%. Furthermore, *Medal Rates* across nearly all models are near zero, and the *Median Rate* peaks at only 14.3%. Among all evaluated models, **GPT-5.1** with high reasoning effort performs the best; we consistently observe that increasing reasoning effort for GPT-5.1 leads to substantial gains across all prediction benchmarks.

### 5.3. Analysis

#### Finding 1: A persistent scientific-domain gap remains even for frontier closed-source models.

Despite strong performance on general-purpose benchmarks (Table 2), all models substantially underperform on the DS-BIO suite (Table 3), which demands bioinformatics workflows and biologically grounded task interpretation (e.g., specialized libraries and modality-specific preprocessing). While small models exhibit a broader mix of planning/statistical failures, the dominant bottleneck for frontier models in realistic scientific analyses is domain grounding.

| Model | QRData-Verified (%) | DABStep-easy (%) | DABStep-hard (%) | DAEval-Verified (%) |
|---|---|---|---|---|
| **Proprietary Models** | | | | |
| GPT-5.1 (high) | 60.16 | 73.61 | 13.23 | 89.50 |
| GPT-5.1 (none) | 58.96 | 70.83 | 11.9 | 87.85 |
| GPT-5 (medium) | 61.75 | 75.00 | 28.31 | 89.50 |
| GPT-4o | 60.24 | 73.61 | 7.41 | 92.26 |
| Claude Sonnet 4.5 | 61.35 | **83.33** | **37.04** | 91.71 |
| Claude Sonnet 4 | 59.06 | 81.94 | 31.75 | 90.91 |
| **Open-sourced Models** | | | | |
| Qwen3 235B Instruct | 54.18 | 73.61 | 17.46 | 85.08 |
| Qwen3-Coder 480B | 54.72 | 75.00 | 14.29 | 90.61 |
| Kimi K2 Instruct | **63.68** | 77.78 | 28.84 | **92.82** |
| GPT-OSS-120B | 47.95 | 70.83 | 7.94 | 84.53 |
| Deepseek-v3.1 | 57.37 | 76.39 | 21.96 | 82.32 |
| Qwen2.5-7B-Instruct | 35.04 | 47.22 | 2.38 | 50.56 |
| Qwen3-4B-Instruct | 45.27 | 58.33 | 2.9 | 64.47 |

*Table 2.* Accuracy performance comparison across standardized general data analysis datasets.

| Model | Overall (%) | Single-Cell Biology (%) | Genetics (%) | Spatial Transcriptomics (%) |
|---|---|---|---|---|
| **Closed-sourced Models** | | | | |
| GPT-5.1 (high) | 38.89 | 45.45 | 28.57 | 28.57 |
| GPT-5.1 (none) | 37.78 | 38.18 | 33.33 | 42.86 |
| GPT-5 (medium) | 32.22 | 34.48 | 33.33 | 18.18 |
| GPT-4o | 33.33 | 43.64 | 4.76 | 35.71 |
| Claude Sonnet 4.5 | 42.22 | 47.27 | 33.33 | 35.71 |
| Claude Sonnet 4 | 36.67 | 38.18 | 33.33 | 35.71 |
| **Open-sourced Models** | | | | |
| Qwen3 235B Instruct | 38.89 | 41.82 | 42.86 | 21.43 |
| Qwen3-Coder 480B | 34.44 | 36.36 | 28.57 | 35.71 |
| Kimi K2 Instruct | **43.33** | 45.45 | 42.86 | 35.71 |
| GPT-OSS-120B | 25.56 | 29.09 | 14.29 | 28.57 |
| Deepseek-v3.1 | 40.00 | 43.64 | 38.10 | 28.57 |
| Qwen2.5-7B-Instruct | 4.44 | 5.45 | 4.34 | 7.14 |
| Qwen3-4B-Instruct | 6.67 | 7.27 | 4.76 | 7.14 |

*Table 3.* Accuracy Performance comparison on DSBIO tasks.

Error breakdowns in Figure 6 further indicate a qualitative shift in failure modes for frontier models. More details are deferred to Appendix D.3. On general analysis tasks, failures are largely attributable to statistical-knowledge and planning issues; however, on DSBIO, domain-grounding errors dominate across all models (85–96% of sampled failures), with representative examples provided in Appendix F.1. Our detailed analysis indicates that these biological grounding failures largely arise from two sources. First, agents often struggle to robustly interpret complex queries together with dataset metadata in the intended biological context. Since DSBIO targets real-world, high-dimensional bioinformatics datasets from published studies, exploratory probing can surface unexpected signals that require specialized biological context; when this happens, agents frequently deviate from their initial plan and resort to trial-and-error reasoning with insufficient domain knowledge (see example in F.1), ultimately producing incorrect answers. Second, agents exhibit limited familiarity with common bioinformatics methods and library usage. They may attempt to reimplement sophisticated algorithms

from scratch rather than leveraging existing functions and libraries provided in the environment, and they often mishandle domain-specific edge cases intrinsic to biological data (e.g., sparsity), leading to missing steps or incorrect preprocessing and downstream analysis.

**Finding 2: Models exhibit persistent behavioral limitations: Technical Constraints and Simplicity Bias.**

Beyond domain-specific knowledge gaps, our evaluation identifies technical constraints that hamper model autonomy. These include **Environment Access Restrictions** (e.g., inability to install libraries or timeouts during large-scale training) and **API Incompatibilities**, manifested as version-specific errors such as hallucinating deprecated keyword arguments (e.g., *early_stopping_rounds* in LightGBM).

However, these mechanical failures compound a more systemic issue: a simplicity bias. As shown in Table 4, model exhibit a large delta between Valid (valid submission generation) and the above-median rate (competitive method). This gap is driven by **Low-Effort Heuristics**, where LLMs optimize for the path of least resistance—such as adopting

| | MLEBench-lite | | | DSPredict-Hard (Private) | | | DSPredict-Easy (Private) | | |
|---|---|---|---|---|---|---|---|---|---|
| Model | Valid | Medal | Median | Valid | Medal | Median | Valid | Percentile | Median |
| GPT-5.1 (high) | 90.91 | **22.73** | **45.45** | **85.7** | **4.8** | **14.3** | **100** | **60.4** | **75** |
| GPT-5.1 (medium) | 90.91 | **22.73** | 31.82 | 81.0 | **4.8** | 7.1 | 91.7 | 55.7 | 63.9 |
| GPT-5.1 (none) | 72.72 | 13.64 | 22.73 | 69.0 | 2.4 | 10.3 | 97.2 | 45.7 | 41.7 |
| GPT-5 (medium) | 77.27 | 9.09 | 27.27 | 52.4 | 0 | 2.4 | 75 | 53.5 | 52.8 |
| Claude Sonnet 4.5 | 86.36 | **22.73** | 36.36 | 71.4 | 0 | 4.8 | **100** | 49 | 52.8 |
| Claude Sonnet 4 | 90.9 | 13.63 | 22.73 | **85.7** | 2.4 | 4.8 | **100** | 44.4 | 36.1 |
| Qwen3-Coder 480B | **100.0** | 9.09 | 22.72 | 66.7 | 2.4 | 5.9 | 86.5 | 42.9 | 33.3 |
| Qwen3 235B Instruct | 81.82 | 4.55 | 13.64 | 64.3 | 2.4 | 2.4 | 97.2 | 42.9 | 33.3 |
| Kimi K2 Instruct | 86.37 | 13.64 | 27.27 | 69 | 0 | 0 | 97.2 | 43.9 | 41.7 |
| Deepseek v3.1 | 86.37 | 13.64 | 27.27 | 76.2 | 2.4 | 7.1 | 86.8 | 30.9 | 7.9 |
| Qwen3-4B-Instruct | 50.0 | 4.55 | 9.09 | 40.5 | 0 | 0 | 67.6 | 28.2 | 5.4 |
| Qwen2.5-7B-Instruct | 0 | 0 | 0 | 4.8 | 0 | 0 | 35.1 | 17.5 | 0 |

*Table 4.* Performance comparison across prediction tasks. For DSPREDICT, results are reported on private test set. We report Valid Submission Rate (*Valid*), Above Median Rate (*Median*), and Any Medal Rate (*Medal*); for DSPREDICT-EASY, we report Percentile rank instead of Medal. Agent is given 10 hours total, and 2 hours per action. Details of metrics and evaluation are deferred to Appendix G.3.

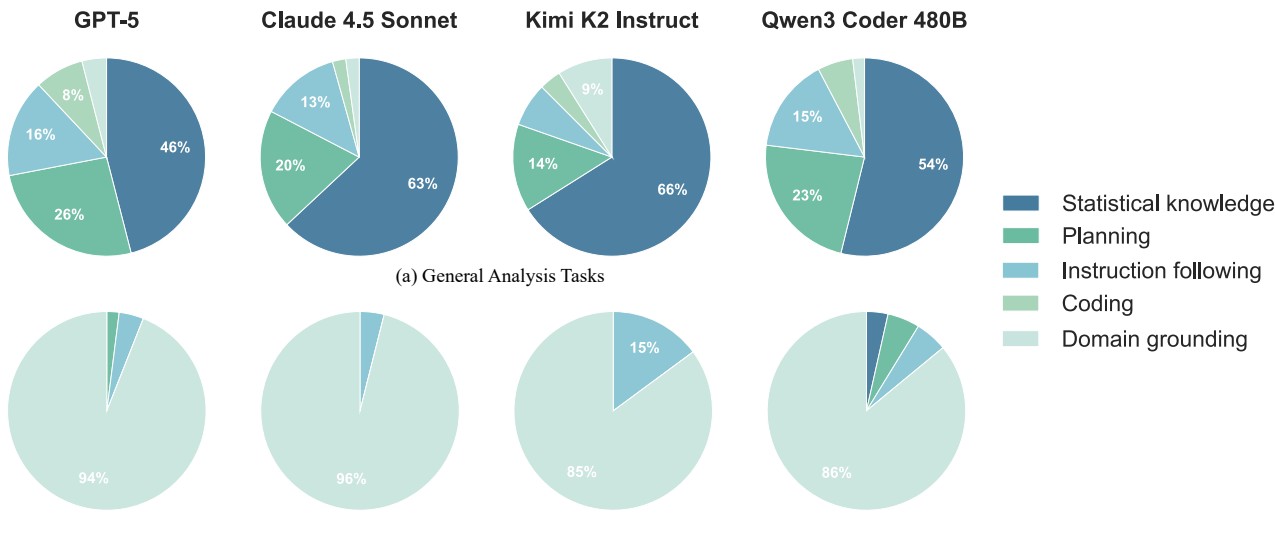

(a) General Analysis Tasks

- Statistical knowledge
- Planning
- Instruction following
- Coding
- Domain grounding

(b) Scientific Analysis Tasks

*Figure 6.* **Error type breakdowns for four LLMs** on (a) general analysis tasks (QRDATA and DAEVAL) and (b) scientific analysis tasks (DSBIO). For each model and task family, we uniformly sample 50 failed trajectories and manually assign a primary error category (definitions in Appendix D.3; representative cases in Appendix F.1). A key shift emerges: while failures on general tasks are dominated by statistical knowledge and planning issues, failures on DSBIO are overwhelmingly driven by domain-grounding errors (85–96% across models).

| Model | QRData-Verified (%) | DABStep-easy (%) | DABStep-hard (%) | DAEval-Verified (%) | DSBIO (%) |
|---|---|---|---|---|---|
| GPT-4o | 60.24 | 73.61 | 7.41 | 92.26 | 33.33 |
| Claude Sonnet 4.5 | 61.35 | **83.33** | **37.04** | 91.71 | 42.22 |
| Claude Sonnet 4 | 59.06 | 81.94 | 31.75 | 90.91 | 36.67 |
| Qwen3-Coder 480B | 54.72 | 75.00 | 14.29 | 90.61 | 34.44 |
| Kimi K2 Instruct | **63.68** | 77.78 | 28.84 | **92.82** | **43.33** |
| Qwen2.5-7B-Instruct | 35.04 | 47.22 | 2.38 | 50.56 | 5.56 |
| Datamind-7B | 49.00 | 68.06 | 2.38 | 85.79 | 15.56 |
| Qwen3-4B-Instruct | 45.27 | 58.33 | 2.9 | 64.47 | 6.67 |
| Jupyter Agent Qwen3 4B | - | 70.80* | 3.4* | - | - |
| **Qwen3-4B-DSGym-SFT-2k** | 59.36 | 77.78 | 33.07 | 86.19 | 21.11 |

*Table 5.* Accuracy performance comparison across data analysis tasks. * means we directly report the numbers in the original report.

a median-based baseline—rather than attempting rigorous, image-based modeling.

Ultimately, these three factors—environmental blocks, API friction, and internal heuristics—collectively drive the simplicity bias. When models encounter technical resistance (environment or API errors), their preference for minimizing trajectory length leads them to abandon complex strategies

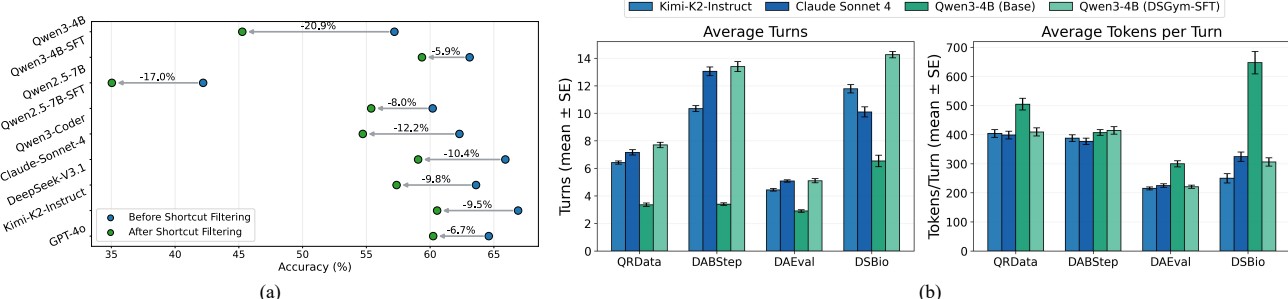

*Figure 7.* (a) **Accuracy on the same error-cleaned QRData split *with vs. without* enforcing data dependency.** All models exhibit consistent drops after filtering, indicating that a non-trivial portion of pre-filter performance can be achieved via non-data-grounded shortcuts (e.g., memorization, priors, etc). (b) **Execution-grounded SFT changes agent interaction behavior toward teacher-like trajectories.** Across four datasets, we report the mean±std of the number of turns per trajectory and tokens per turn for two teacher models and a 4B base model before/after DSGym-SFT. DSGym-SFT increases the number of turns while shifting tokens-per-turn toward teacher-like statistics, indicating finer-grained decomposition and more iterative execution.

in favor of superficial, safe analysis. This suggests that frontier models, while proficient at code generation, lack the "skeptical" persistence of expert data scientists, treating the first valid result as ground truth rather than a hypothesis to be improved. More details of our failure analysis can be found in Section D.1.

**Finding 3: Shortcut filtering reveals substantial non-data-dependent solvability; smaller open-weight models are affected most.** As shown in Fig. 7(a), enforcing data dependency consistently decreases accuracy across all evaluated models on the same error-cleaned QRData split (up to ∼21% relative drop). Representative examples of tasks solvable without files are provided in Appendix F.3.

### 5.4. Case Study: Training Data Science Agents via DSGYM

To demonstrate the potential utility of DSGYM for training, we utilize the architecture to generate 2,000 execution-grounded synthetic queries and trajectories for general data analysis tasks, denoted as DSGYM-SFT. Details of the synthetic generation process and more analysis of the trained agent are deferred to Section A. Table 5 shows that DSGYM-SFT yields consistent gains over the Qwen3-4B base model across benchmarks, with particularly large improvements on DABSTEP-HARD. Notably, although DSGYM-SFT is constructed only on *general* data analysis tasks, it also improves performance on the out-of-domain DSBIO benchmark, suggesting that the planning, reasoning, or decomposition-oriented behaviors learned from general analysis can transfer to scientific workflows beyond the training distribution though domain grounding remains a severe bottleneck.

### 6. Related Work

**Benchmarks for Data Science.** Recent work has increasingly focused on evaluating LLMs on data science tasks

beyond basic code generation. Early benchmarks targeted relatively simple, single-step data analysis problems with constrained library usage or interactive notebook settings (Lai et al., 2023; Yin et al., 2023). More recent efforts aim to capture realistic data science workflows, incorporating multi-step reasoning, iterative execution, debugging, visualization, and domain knowledge (Huang et al., 2024; Hu et al., 2024; Majumder et al., 2024; Zhang et al., 2025; Lu et al., 2025). These benchmarks span diverse task types, including statistical reasoning, data exploration, visualization, long-context interaction, and end-to-end machine learning pipelines (Liu et al., 2024; Yang et al., 2024; Egg et al., 2025; Jing et al., 2024; Gu et al., 2024; Chan et al., 2024). In contrast to task-specific benchmarks, DSGYM emphasizes a standardized gym-style environment that unifies heterogeneous data science tasks and interfaces, enabling reproducible training and evaluation of agent systems. A more detailed discussion is provided in Appendix B.

### 7. Conclusion

We introduce DSGYM, a standardized, extensible framework for evaluating data science agents in stateful, isolated execution environments. DSGYM unifies heterogeneous benchmarks under a single abstraction and enables reproducible, end-to-end measurement of LLMs as data science agents. Crucially, we challenge the assumption that file-grounded benchmarks necessarily test data-dependent reasoning and provide tooling to mitigate prompt-only shortcut solvability. We release DSGYM-TASKS: audited and standardized analysis benchmarks with shortcut filtering, and two novel suites: DSBIO for domain-grounded scientific analysis and DSPREDICT for realistic end-to-end modeling. We hope DSGYM serves as a live, auditable testbed that evolves with scientific practice while providing a moving yet reproducible target for evaluating and advancing LLM-based data science agents.

## Impact Statement

This paper presents DSGym, a benchmark and execution framework for evaluating LLM agents on data science and scientific analysis tasks. The primary goal of this work is to advance the methodology for evaluating agentic systems in a way that is more faithful to real-world data-dependent workflows. We anticipate that improved evaluation practices can have positive downstream impact by enabling more reliable development, comparison, and deployment of LLM-based agents in scientific and data-driven settings. By highlighting failure modes such as shortcut solvability and lack of data grounding, DSGym may help prevent overestimation of agent capabilities and reduce the risk of deploying systems that appear competent but fail under realistic conditions.

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

# Appendix

## A. Demonstration: Training Data Science Agents via DSGYM

Beyond evaluation, DSGYM also enables research on training data science agents with different algorithms such as supervised finetuning, curriculum learning, and reinforcement learning with the help of its distributed environment, standardized datasets and trajectory recording infrastructure. In this section, we demonstrate how DSGYM can be used to construct high-quality synthetic training data through synthetic query construction and trajectory generation. These procedures provide a practical example of leveraging the DSGYM environment for agent training without human intervention.

### A.1. Execution-Grounded Data Synthesis

We adopt a multi-stage process grounded in execution at every step to synthesize training data.

**Stage 1: Exploratory Query Generation.** To ensure generated questions are grounded in reality, we employ an "Explore-and-Validate" method. We utilize the default agent scaffold in DSGYM for generating synthetic queries. The agent will be given an example query without ground-truth, context information, and dataset files, and then the agent can interact with the environment to come up with semantically distinct questions. The agent is instructed to avoid trivial rephrasings and to design realistic tasks that can be solved through executable analysis. Critically, the agent is required to output not just the question, but also a reference Answer and strict answer format guidelines. To fulfill this requirement, the generator agent must interact with the environment—loading data, inspecting schemas, and actually *solving* its own proposed query via code execution. This self-validation step ensures that every synthesized query is feasible.

**Stage 2: Trajectory Sampling.** Once the valid queries and their reference answers are obtained, we generate diverse solution paths. We instantiate a fresh DSGYM environment for each query and use the default agent scaffold to generate $K$ independent candidate trajectories with temperature $T = 0.8$.

**Stage 3: Joint Query-Trajectory Validation.** We employ an LLM-based Judge to evaluate the Query-Trajectory pair as a coherent unit. Unlike simple answer matching, the judge evaluates the query and the whole trajectory using six execution-aware criteria:

- **Query Clarity and Feasibility:** Is the query clearly-defined, unambiguous and realistically solvable?
- **Educational Value**: Does the query have learning value and sufficient complexity?
- **Exploratory Competence:** Does the trajectory perform sufficient data exploration?
- **Execution Robustness:** Are code blocks runnable? If errors occurred, did the agent successfully debug and recover?
- **Task Alignment:** Does the executed logic actually address the specific intent of the query?
- **Answer Plausibility:** Is the derived answer consistently supported by the final execution outputs and consistent with the reference answer?

After this quality filtering, we apply a lightweight Diversity Filter based on semantic similarity to discard synthesized queries that are trivial rephrasings of the original seed example.

**Applicability to Existing Benchmarks.** While described above as a full synthesis pipeline, the Trajectory Sampling and Verification stages (Stages 2 & 3) function as a modular subsystem. They can be applied directly to *existing tasks* to distill high-quality, execution-verified reasoning traces for SFT.

### A.2. Case Study: The DSGYM-SFT Dataset

To demonstrate the utility of this pipeline, we constructed a demonstration training corpus. Starting from a seed subset of QRDATA and DABSTEP, we prompted agents to explore the datasets and generate 3,700 synthetic query candidates. These were re-executed to obtain full reasoning traces. After applying our Joint Query-Trajectory Filtering, we curated **2,000 high-quality pairs**. This dataset, denoted as DSGYM-SFT, represents a fully synthetic, execution-verified instruction tuning corpus.

This example illustrates how DSGYM transforms from a purely evaluative benchmark into a closed-loop training ecosystem, enabling scalable generation, assessment, and refinement of data-science agents through realistic, executable analytical tasks.

### A.3. Experiments

Table 5 shows that a 4B model fine-tuned on DSGYM-SFT attains competitive performance relative to substantially larger baselines, illustrating the potential of execution-grounded synthesis for data-efficient improvement.

**Data-efficient gains on analysis tasks.** Fine-tuning on DSGYM-SFT yields consistent gains over the Qwen3-4B base model across benchmarks, with particularly large improvements on DABSTEP-HARD. Notably, although DSGYM-SFT is constructed only on *general* data analysis tasks, it also improves performance on the out-of-domain DSBIO benchmark, suggesting that the planning, reasoning, or decomposition-oriented behaviors learned from general analysis can transfer to scientific workflows beyond the training distribution.

**More structured interaction behavior.** Beyond accuracy, DSGym-SFT also changes how agents interact with the environment: as shown in Fig. 7(b), SFT increases depth of exploration, promotes finer-grained decomposition, and encourages iterative execution, which likely contributes to improved performance on complex workflows in DABSTEP-HARD and DSBIO.

**Less reliance on shortcut solvability.** Fig. 7(a) indicates that smaller open-weight models experience the largest performance degradations when shortcut solutions are removed. In contrast, DSGYM-SFT models exhibit substantially smaller drops, suggesting improved robustness to shortcut-based answering.

## B. Related Works

### B.1. Benchmarks for Data Science

Assessing LLMs' data science capabilities has been actively studied in recent years. Early research focused on relatively simple code-generation tasks; for instance, (Lai et al., 2023) investigated introductory-to-intermediate data analysis problems restricted to the use of seven commonly used Python libraries (e.g., Numpy (Harris et al., 2020) and Pandas (pandas development team, 2020)), and (Yin et al., 2023) explored problems of similar difficulty in interactive data science notebook settings. Although these benchmarks support fast and automated evaluation, their simplicity limits to capture multi-step and interactive agent behaviors. This limitation has motivated subsequent work to incorporate more realistic and challenging components, including iterative reasoning/planning, statistics/domain knowledge, repeated code execution, and debugging within an interactive environment (Huang et al., 2024; Hu et al., 2024; Majumder et al., 2024; Zhang et al., 2025; Lu et al., 2025). As a few representative examples, in data analysis tasks, (Liu et al., 2024) curated reasoning tasks from statistics textbooks that require both data input/output processes and data exploration, (Yang et al., 2024) introduced a benchmark framework for evaluating LLMs' visualization ability, (Egg et al., 2025) examined financial data analysis involving multi-step reasoning over heterogeneous data sources, (Jing et al., 2024) studied agent behavior under long-context settings, and (Gu et al., 2024) considered open-ended data science questions collected from scientific literature. In predictive modeling tasks, (Chan et al., 2024) curated 75 Kaggle competitions and examined how well LLM-based agents handle end-to-end ML engineering tasks. DSGYM focuses more on providing a gym environment tailored to data science tasks, standardizing heterogeneous data and model interfaces.

### B.2. Agents for Data Science

Alongside benchmarks, a growing body of work studies agent scaffolds; how to structure agents to handle complex data science workflows (Rahman et al., 2025). Many early approaches, including (Huang et al., 2023; Hu et al., 2024), rely on a single linear execution trace as variations of ReAct or CodeAct (Yao et al., 2022; Wang et al., 2024b) and have shown promising abilities of these agents. Recently, (Jiang et al., 2025) improved upon this paradigm by representing candidate solutions as nodes in a tree. This tree representation enables the agent to explore multiple candidate solutions in parallel, backtrack from suboptimal trajectories, and refine the final solution. (Yang et al., 2025b) further enhanced this scaffold with more sophisticated planning/reasoning modules, in which the agent generates ideas and verifies them multiple times before implementation. This approach has been shown to be effective for developing predictive models and achieves competitive performance on MLE-bench (Chan et al., 2024). Overall, the agent performance highly depends on how the system structures iteration and reasoning; effective agents explicitly conduct multi-step search over hypotheses, candidate solutions, and evaluation feedback.

Beyond scaffold-based approaches, many studies have explored the design of data science agents from multiple perspectives, including environment modeling, agent coordination, and task representation. (You et al., 2025) developed an agent that can

interact on a sequence of markdown or executable code cells in Jupyter Notebook environments. (Li et al., 2024) considered a multi-agent system capable of completing end-to-end data science workflows, ranging from data preprocessing to report generation. From data-structural perspectives, (Hong et al., 2025) considered representing a data science task as a graph, dynamically decomposing the main task into dependent subtasks and revising the graph as new evidence or constraints appear. The main goal of DSGYM is to provide an easy-to-use, standardized system that supports reproducible training and evaluation of agent systems, making it simple for these agents to be adopted and assessed.

## C. Discussion and Limitations

Our findings highlight both the opportunities and ongoing challenges in leveraging LLMs as agents for automated data science. We now discuss the several avenues for improvement:

- **Extending to RL.** A key advantage of DSGYM is its distributed, containerized, stateful execution, which naturally supports interactive optimization of agent policies. This makes DSGYM a suitable environment for studying RL-style training and evaluation across multiple data science datasets. However, two challenges remain central: *training signal design* and *data and task coverage*. Existing data science trajectories are limited in scale, uneven in quality, and often underrepresent domain-specific scientific workflows. Moreover, providing informative credit assignment under sparse, long-horizon rewards remains an open problem. DSGYM exposes these challenges in a controlled setting, enabling systematic investigation of reward design and verification-based filtering.

- **Deepening Scientific Grounding.** Our analysis on DSBIO shows that generalist models struggle with domain-specific ontologies, data modalities, and tooling, with the gap particularly pronounced for smaller models. Two complementary directions may help address this limitation. Tool-oriented abstractions can reduce avoidable workflow errors by exposing robust domain primitives, while domain-adaptive learning (e.g., continued pretraining or finetuning on scientific corpora and verified analysis traces) may be necessary to improve conceptual grounding and method selection. Expanding to additional scientific domains (e.g., chemistry, materials science, or astronomy) is also important, not merely to increase task diversity, but to probe qualitatively different forms of domain knowledge under a standardized evaluation interface.

- **Extending beyond Python.** DSGYM is currently Python-centric, but its manager-worker architecture is designed to be language-agnostic. Supporting other ecosystems such as R would mainly require adding a containerized worker image with the appropriate runtime and kernel support (e.g., R with IRKernel), and using task metadata to route tasks to the corresponding backend. This would allow DSGYM to cover mixed-language scientific workflows while preserving isolated execution, resource control, dataset mounting, and reproducible evaluation.

- **Deterministic evaluation and open-ended discovery.** We intentionally prioritize reproducibility through strict data dependency and deterministic evaluation metrics. However, many real-world scientific workflows are open-ended, involving stochastic outcomes, visualization, or multiple valid interpretations. DSGYM currently does not cover such settings, including visualization-centric or exploratory tasks. Extending evaluation beyond deterministic regimes remains challenging and will likely require reliable validation mechanisms grounded in execution traces, such as carefully controlled LLM-based judges.

- **DSGYM as a live testbed.** We envision DSGYM as a living testbed that evolves with scientific tooling and emerging evaluation needs, complementing static benchmarks that are prone to memorization and rapid saturation. This live-but-auditable design supports reproducible measurement, systematic ablations, and principled tracking of progress over time.

- **Distinction of DSPREDICT-HARD from MLE-Bench.** The primary distinction between our curated DSPREDICT-HARD collection and MLE-BENCH LITE lies in the recency, accessibility, and domain coverage. Our dataset focuses on newer Kaggle challenges. The oldest from 2017 and several from 2024 to 2025, thereby reducing the likelihood of data leaks and ensuring that tasks better reflect contemporary data science practices. In addition, we include only competitions that still accept submissions, allowing us to obtain official leaderboard scores and ensure accurate, up-to-date evaluation. Finally, DSPREDICT-HARD spans a broader and more heterogeneous set of application domains (e.g., sensor signal, business, sports, and time-series forecasting tasks, see more in Table S4), ensuring that performance reflects generalizable data science competence.

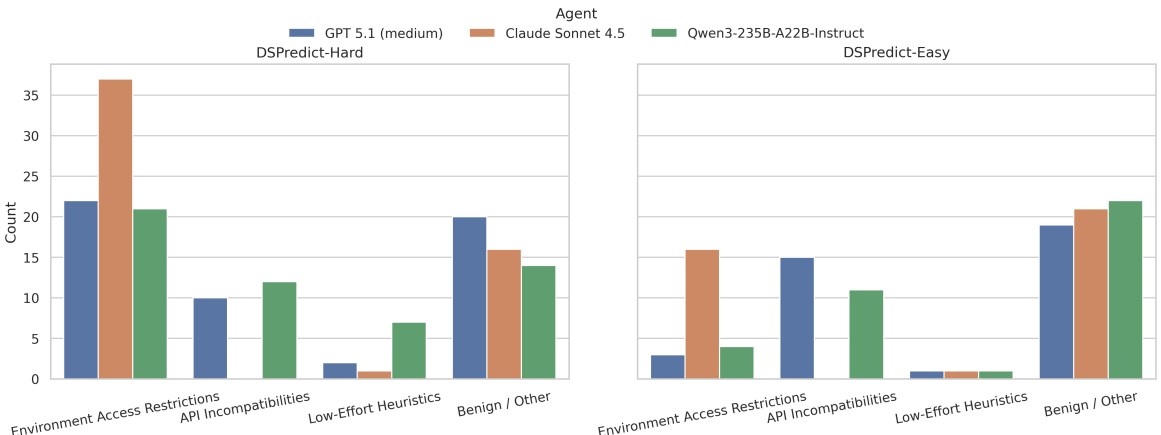

*Figure S1.* Failure modes for agents on DSPREDICT-HARD. Three models are annotated with four categories.

## D. Additional Evaluation and Analysis Results

### D.1. DSPredict Failure Mode Analysis

To better understand the operational bottlenecks of autonomous data science agents, we conducted a taxonomy of failure modes across the DSPREDICT-HARD and DSPREDICT-EASY benchmarks. Figure S1 illustrates the distribution of error categories for three state-of-the-art models: GPT 5.1 (medium), Claude Sonnet 4.5, and Qwen3-235B-A22B-Instruct.

We classified agent failures into four primary categories:

- **Environment Access Restrictions**: Failures resulting from timeouts or attempts to install unauthorized external libraries.
- **API Incompatibilities**: Errors stemming from version mismatches, such as the hallucination of deprecated arguments (e.g., early_stopping_rounds in LightGBM).
- **Low-Effort Heuristics**: Cases where the agent defaulted to simplistic baselines (e.g., "median pose") rather than attempting robust modeling.
- **Benign / Other**: Successful runs or outliers not fitting the primary failure definitions.

The results highlight a trade-off between code complexity and execution robustness. In the DSPREDICT-HARD setting, Claude Sonnet 4.5 exhibits the highest frequency of Environment Access Restrictions ($N = 37$), significantly outpacing other models. This suggests that while Claude generates sophisticated solutions, it frequently misjudges runtime constraints (e.g., internet access or time out). However, it demonstrates near-zero API Incompatibilities, indicating superior internalization of library standards compared to GPT 5.1 and Qwen3, which struggle with version-specific syntax.

Furthermore, task difficulty influences agent laziness. Qwen3 shows a notable increase in Low-Effort Heuristics on the hard benchmark, implying a tendency to prioritize path-of-least-resistance baselines (e.g., median pose) when facing high-complexity modeling challenges. Conversely, DSPREDICT-EASY shows a flatter distribution with higher Benign completion rates, confirming that infrastructure constraints become the primary bottleneck only as task complexity scales.

### D.2. DSPREDICT Loop Agent Result

Table S1 illustrates the performance trajectory of the LLM agents on the DSPREDICT-HARD Public and Private benchmarks across three sequential iterations. Specifically, each agent will attempt each competition three times. Each time the agent will be given reflections from last attempt for improvement to help its next attempt. Under a strict computational budget of 10 hours total, with a 4.5-hour timeout enforced per action, the agents demonstrate a monotonic increase in accuracy from Loop 1 to Loop 3. This positive trend substantiates the efficacy of the autonomous reflection module, which allows the agent to analyze execution logs from preceding attempts and refine its policy for subsequent runs.

While overall performance improves on average, the effectiveness of the iterative looping mechanism varies substantially across data splits and model architectures. In particular, gains on the Private benchmark are notably smaller than those observed on the Public split, suggesting that the benefits of self-reflection may partly depend on test-set–specific patterns or

| Model / Scaffold | DSPredict-Hard (Public) | | | DSPredict-Hard (Private) | | |
|---|---|---|---|---|---|---|
| | Valid | Medal | Median | Valid | Medal | Median |
| **GPT-5.1 (medium)** | | | | | | |
| Loop 1 | 81.5 | 3.7 | 14.8 | 78.6 | **4.8** | 9.5 |
| Loop 2 | 79.6 | 5.6 | 16.7 | 78.6 | 2.4 | 9.5 |
| Loop 3 | **90.7** | **9.3** | **27.8** | **90.5** | **4.8** | **11.9** |
| **Claude 4.5 Sonnet** | | | | | | |
| Loop 1 | 66.7 | 7.2 | 13.0 | 69.0 | **2.4** | **14.3** |
| Loop 2 | 70.4 | **7.4** | 13.0 | 73.8 | **2.4** | **14.3** |
| Loop 3 | **77.8** | **7.4** | **14.8** | **78.6** | **2.4** | **14.3** |
| **Qwen3-Coder-480B-A35B** | | | | | | |
| Loop 1 | **68.5** | 1.9 | **9.3** | **76.2** | 0.0 | 0.0 |
| Loop 2 | **68.5** | 1.9 | 7.4 | **76.2** | 0.0 | **2.4** |
| Loop 3 | 61.6 | **3.7** | **9.3** | 59.5 | 0.0 | **2.4** |
| **Qwen3-235B-A22B** | | | | | | |
| Loop 1 | 59.3 | **3.7** | 7.4 | 54.8 | 0.0 | **2.4** |
| Loop 2 | 57.4 | **3.7** | 7.4 | 57.1 | 0.0 | **2.4** |
| Loop 3 | **61.1** | **3.7** | **9.3** | **59.5** | 0.0 | **2.4** |
| **Kimi-K2-Thinking** | | | | | | |
| Loop 1 | **68.5** | **3.7** | 7.4 | 63.6 | **2.4** | **6.8** |
| Loop 2 | 63.0 | **3.7** | 7.4 | 61.9 | **2.4** | 4.8 |
| Loop 3 | **68.5** | **3.7** | **9.3** | 64.3 | **2.4** | 4.8 |
| **Deepseek v3.1** | | | | | | |
| Loop 1 | 59.3 | **3.7** | 5.6 | 64.3 | 0.0 | **4.8** |
| Loop 2 | **66.7** | 0.0 | **7.4** | **69.0** | 0.0 | 2.4 |
| Loop 3 | **66.7** | 0.0 | **7.4** | **69.0** | 0.0 | 2.4 |

*Table S1*. Performance progression of LLM agents on DSPredict-Hard Public and Private benchmarks. Each agent performs three attempts per competition, incorporating reflections from prior attempts. All agents operated under a 10-hour total time limit with a 4.5-hour execution timeout per action. The results demonstrate a progressive performance gain across successive iterations (Loops 1–3), where each subsequent run is informed by an autonomous reflection on the preceding attempt. All other setup is the same as Table 4.

may not fully generalize to the distinct distribution of the held-out private data. Moreover, the ability to self-correct appears closely tied to the baseline reasoning capacity of the model. Stronger models such as GPT-5.1 and Claude 4.5 Sonnet exhibit consistent, monotonic improvements across iterations (e.g., GPT-5.1 improving from 14.8% to 27.8%), whereas weaker models, including Qwen3-Coder and Kimi-K2, show diminishing returns or even performance degradation in later loops. These results suggest that, in the absence of sufficient reasoning capability, autonomous reflection may amplify noise rather than reliably correct errors.

### D.3. Analysis for Data Analysis Tasks

In order to analyze error patterns across models and domains, we conduct a manual failure analysis on unsuccessful trajectories (Figure 6). We use QRDATA and DAEVAL to represent general data-analysis tasks, and DSGYM-BIO to represent scientific analysis tasks. For each model and each task family, we uniformly sample 50 failed trajectories. Each trajectory is independently annotated by two annotators with a single *primary* error type (the earliest/root cause that makes the trajectory fail). Disagreements are resolved through discussion to reach a consensus label. The definitions of each error types are defined as follows:

- **Domain grounding error:** The trajectory fails due to incorrect *domain-specific* understanding or choices that require specialized knowledge to validate (e.g., misinterpreting domain concepts, modality-specific data structures, or scientific

principles; selecting an inappropriate domain-specific library/tool/method for the task). Purely general reasoning issues or generic programming mistakes do *not* qualify as domain grounding errors.

- **Statistical knowledge error:** The trajectory applies an incorrect statistical/mathematical procedure or draws an invalid inference from results, assuming the task is otherwise correctly understood.

- **Planning error:** Poor task decomposition, incorrect approach selection, or flawed reasoning strategy that are domain-agnostic.

- **Instruction following error:** Not adhering to task requirements or format specifications.

- **Coding error:** Programming mistakes, syntax errors, or incorrect implementation.

# E. Additional Details of DSGYM-TASKS

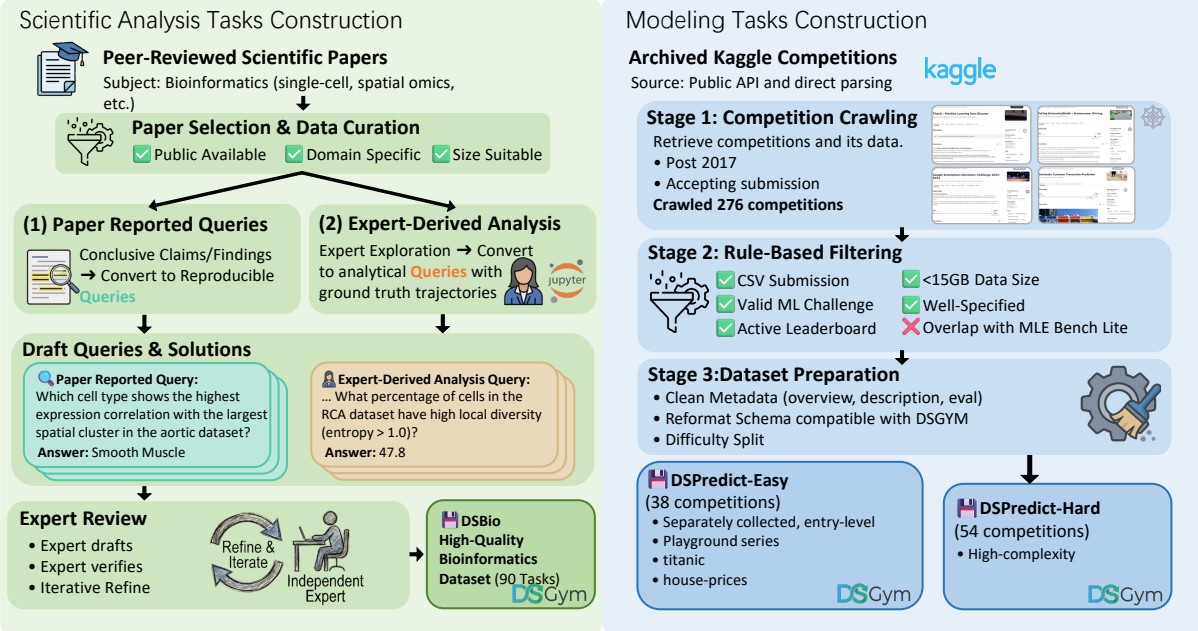

*Figure S2.* **Dataset Construction Pipeline.** Our data construction pipeline curates domain-specific scientific tasks from academic literature and aggregates real-world predictive modeling challenges from Kaggle competitions.

### E.1. Additional Details of Refinement of Existing Benchmarks

Below we summarize the refined subsets:

- **DAEval-Verified.** We remove samples lacking ground truths or containing misaligned question–answer pairs, refine answer-format guidelines (e.g., rounding precision inconsistencies) to match with the ground-truths, and correct typographical errors. The resulting dataset, categorized as data analysis tasks, provides short analytical queries that serve as a basic, general-purpose evaluation of data handling and statistical competence.

- **QRData-Verified.** We remove invalid multiple-choice queries with duplicate or ambiguous choices. This dataset focuses on statistical and causal reasoning over tabular data and belongs to the data analysis category.

- **DABStep.** DABStep comprises financial multi-step analytical queries that require reasoning across multiple data files.

- **MLEBench-Lite.** We integrate MLEBENCH-LITE as a canonical data prediction benchmark within DSGYM, ensuring full compatibility with our unified environment and metric registry.

Here we provide examples of the tasks that we filter.

**Question**: *Compute the proportion of patients in the treatment group who had a stroke by the end of their first year.* (Keywords: Statistics, Probability )

**Question**: *Which cause-and-effect relationship is more likely?* (Keywords: Causal Analysis)

**Question**: *Estimate the standard error of the point estimate from the Ebola survey.* (Keywords: Statistics, Sampling )

**DAEval**

**Question**: *Are there any outliers in the average wait time for callers before being answered by an agent?* (Keywords: Outlier Detection)

**Question**: *Calculate the average closing price for each month and year combination.* (Keywords: Data Preprocessing, Summary Statistics)

**Question**: *Is there a correlation between the maximum storm category achieved by a storm and the recorded damage in USD?* (Keywords: Correlation Analysis)

**QRData**

**Question**: *Which developmental transcription factors along with ISL1 and MEF2C shows significant motif enrichment in genes differentially expressed between ascending and descending fibroblasts?* (Keywords: Single-Cell Biology, Enrichment Analysis)

**Question**: *Calculate the nearest neighbor enrichment for cell type pairs using k=10 neighbors. Which cell type pair shows the strongest spatial co-localization based on the enrichment score?* (Keywords: Spatial Colocalization Analysis, Nearest Neighbor Enrichment)

**DSBio**

**Question**: *What are the applicable fee IDs for Belles_cookbook_store in 2023?* (Keywords: Rule-based Analysis, Data Integration, Filtering)

**Question**: *For the 200th of the year 2023, what is the total fees (in euros) that Crossfit_Hanna should pay?* (Keywords: Data Aggregation, Filtering, Rule-based Calculation)

**DABStep**

*Figure S3.* **Example questions across data science benchmarks.** Existing datasets such as QRDATA, DAEVAL, and DABSTEP mainly target general or applied data-science operations. DSGYM complements these with new domain-specific scientific tasks (e.g., bioinformatics) that require specialized workflows and terminology.

---

**Dataset:** QRData

**Task:** Which cause-and-effect relationship is more likely? Please answer with A, B, or C.

    A. L tibia pain causes L tibia pain
    B. L tibia pain causes L tibia pain
    C. No causal relationship exists

**Provided Answer:** C

**Issue Identified:** The answer choices contain duplicated options (A and B are identical), making the task ill-defined.

**Action Taken:** This task is filtered out during dataset refinement due to invalid answer options.

---

**Dataset:** DAEval

**Task:** Is there a significant difference in the total number of vaccinations administered per hundred people between countries that use different vaccines?

Constraints:
Only consider countries using Pfizer/BioNTech, Moderna, Oxford/AstraZeneca, and Johnson&Johnson/Janssen.
The country must have data without null values in the column of total vaccinations per hundred people.
Use One-Way Analysis of Variance (ANOVA) to test if there's significant difference among different vaccine groups.
Consider the differences among vaccine groups to be significant if the p-value is less than 0.05.

Answer Format: {
@significance_of_difference[significance]
@p_value[p_value]
Where 'significance' is a string that can either be 'yes' or 'no' based on the conditions specified in the constraints.
Where 'p_value' is a number between 0 and 1, rounded to four decimal places.
}

**Expected Output Format:**

```
@significance_of_difference[significance]
@p_value[p_value]
```

**Provided Answer:** `[['significance_of_difference', 'no']]`

**Issue Identified:** The required `p_value` field is missing from the provided answer, despite being explicitly required by the task format.

**Action Taken:** This task is filtered out during dataset refinement due to incomplete ground-truth annotation.

---

Here we provide examples of the tasks that we refine.

## E.2. More details about DSB𝗂o

DSGym-bio is a curated benchmark of **90** data-science questions grounded in publicly available biomedical research datasets. Table S2 summarizes the domain distribution. The benchmark primarily focuses on **single-cell biology** (56/90), reflecting both its prominence in modern bioinformatics and the availability of many high-quality, reasonably sized public datasets that fit our agent environment. We additionally include problems from **genetics** (21/90) and **spatial transcriptomics** (13/90) to broaden coverage across biomedical modalities. Table S3 lists representative research papers used to construct DSGym-bio, along with the number of problems derived from each paper, their domain labels, and the corresponding data sources.

**Task Construction Pipeline.** We select seven papers spanning single-cell omics, spatial omics, multi-omics integration, and human genetics. Papers are chosen only if they provided publicly available datasets of a size suitable for loading and analysis within our sandbox environment, avoiding excessive computational overhead. To ensure both coverage and depth, we construct tasks via two complementary ways:

**(1) Reproduction of Reported Findings.** We identify conclusive claims or quantitative findings reported in the original publications and convert them into executable queries. To ensure compatibility with DSG𝗒𝗆, a query is included only if: (i) it can be answered purely from the provided dataset without visual inspection of figures, (ii) it produces a deterministic numerical or factual output.

**(2) Expert-Derived Follow-Up Analyses.** Domain experts conduct a deep exploratory analysis of each dataset in a Jupyter notebook and design queries that require comprehensive, bottom-up reasoning from raw data, rather than simple information retrieval. We intentionally emphasized analytical difficulty by focusing on tasks involving statistical modeling, multi-dataset integration, and minimal reliance on pre-wrapped, domain-specific software packages.

**Iterative Expert Review.** To ensure the quality of the tasks and address the issue of nondeterminism common in scientific open-ended tasks, we implement an iterated expert verification process:

1. A primary expert who drafts the analysis provides the task query and a 'Gold Notebook' solution.
2. An independent expert reviews the quality and difficulty of the task and attempts to solve the task given only the prompt and data.
3. If both solutions match and the task demonstrates sufficient analytical depth, it is accepted; otherwise, if the task is too simple, ambiguous or not deterministic, it is discarded or refined and re-reviewed until consensus is reached.

**Future Extensions to Other Scientific Domains** While the current release focuses on bioinformatics, this construction pipeline is domain-agnostic and designed to extend to fields such as geoscience, computational chemistry and economics in future .

*Table S2.* Distribution of question domains in the DSGym-bio dataset.

| Category | Count |
|---|---|
| Single-cell biology | 55 |
| Genetics | 21 |
| Spatial transcriptomics | 14 |
| **Total** | **90** |

## E.3. Examples of DSB𝗂o

**Dataset:** DSGym-bio
**Domain:** Single-cell biology
**Task:** Identify co-expression modules in endothelial cells using hierarchical clustering on gene-gene correlation matrix. Using Pearson correlation on the top 500 most variable genes, cut the dendrogram at height 0.7 to define modules. How

*Table S3.* Overview of research papers included in **DSGym-bio**, with their publication venues.

| Paper Title | Problem Count | Domain | Data Source |
|---|---|---|---|
| *Single-Cell Transcriptomic Profiling Identifies Molecular Phenotypes of Newborn Human Lung Cells (Bhattacharya et al., 2024)* | 12 | Single-cell biology | cellxgene |
| *A cell and transcriptome atlas of human arterial vasculature (Zhao et al., 2025)* | 14 | Spatial Transcriptomics | cellxgene |
| *Single-nucleus chromatin accessibility and transcriptomic map of breast tissues of women of diverse genetic ancestry (Bhat-Nakshatri et al., 2024)* | 21 | Single-cell biology | cellxgene |
| *Shared genetic effects on chromatin and gene expression indicate a role for enhancer priming in immune response (Alasoo et al., 2018)* | 9 | Genetics | zenodo |
| *Molecular and functional variation in iPSC-derived sensory neurons (Schwartzentruber et al., 2018)* | 12 | Genetics | EMBL-EBI |
| *Human $\gamma\delta$ T cells in diverse tissues exhibit site-specific maturation dynamics across the life span (Gray et al., 2024)* | 12 | Single-cell biology | cellxgene |
| *Single-cell transcriptome analysis reveals differential nutrient absorption functions in human intestine (Wang et al., 2019)* | 10 | Single-cell biology | cellxgene |

---

many genes belong to the largest co-expression module?
**Answer Guideline:** Answer must be a single numeric value (e.g., 42) with no units or text.
**Ground Truth Answer:** 19

---

**Dataset:** DSGym-bio
**Domain:** Genetics
**Task:** Among response eQTL-caQTL pairs, what fraction shows chromatin QTL activity in naive macrophages before stimulation (fold change > 1.5)? Choose among these options: 20%, 40%, 60%, or 80%.
**Answer Guideline:** Answer must be one of the provided options exactly as shown, case-sensitive. For example: '20%'.
**Ground Truth Answer:** 60%

---

**Dataset:** DSGym-bio
**Domain:** Spatial Transcriptomics
**Task:** Which scRNA-seq cell type shows the highest expression correlation with the largest spatial cluster in the aortic Slide-seqV2 dataset? Hint: look at 'author_cell_type' annotation in metadata.
**Answer Guideline:** Answer must be the exact single cell type name as shown in the metadata (e.g., 'Endothelial'), case-sensitive.
**Ground Truth Answer:** Smooth Muscle

---

### E.4. More details of Data Prediction Tasks

Below we provide the details of construction pipeline for DSPREDICT:

**Stage 1: Competition Crawling.** We first deploy a crawler that retrieves all archived Kaggle competitions through the Kaggle public API. For each competition, the crawler extracts the complete descriptions from web pages and the corresponding data files are automatically downloaded. Given the large number of available Kaggle competitions, we restrict our crawl to those that closed after 2017, and still accept submissions. In total, this stage collected 276 competitions spanning a broad range of challenges across structured data, text, and image modalities.

**Stage 2: Rule-Based Filtering.** Next, we apply a rule-based filtering to ensure that only well-structured, executable data science competitions remain:

1. Format & Size: Submissions must be in CSV format; datasets must be under 15 GB for hardware feasibility.
2. Core Focus: Must be a valid ML challenge (no CTFs or code golf) requiring meaningful engineering and pipeline design.
3. Evaluation: Requires an active leaderboard for quantitative benchmarking.
4. Clarity: Objectives and data structures must be well-specified to support reproducibility.
5. Uniqueness: Minimal overlap with MLE Bench Lite.

A complete rule set is deferred to Section E.5.

**Stage 3: Dataset Preparation.** For each filtered competition, we perform standardized data preparation. Competition metadata (overview, data description, and evaluation details) are cleaned and reformatted into a consistent schema compatible with the DSGYM dataset abstraction (Section 2.1). We further categorize the resulting competitions into two difficulty splits.

We provide the full list of competitions in Table S4.

### E.5. Details of Rule-Based Filtering for DSPREDICT

A more detailed version of our rule-based filtering of Kaggle competitions is shown here.

- Submissions must use CSV format to standardize automated submission handling and evaluation.
- The competition must be a valid machine learning challenge (excluding CTFs and code golf tasks) to ensure relevance to data science modeling rather than puzzle solving or code optimization.
- The dataset size must be under 15 GB to ensure feasible data loading and model training on typical research hardware.
- The competition must have an available leaderboard to enable benchmarking and quantitative comparison of model performance.
- The competition should require meaningful ML or data science engineering effort to solve, ensuring that it tests practical modeling, feature engineering, and pipeline design skills.
- The competition description should be well-specified and solvable, providing clear objectives, evaluation criteria, and data structure to support reproducible experimentation.
- Most of the competitions should not overlap with MLE Bench Lite.

*Table S4.* Competition Dataset Sizes categorized by difficulty and source

| Competition | Data Size | Domain |
|---|---|---|
| **DSPREDICT-EASY** | | |
| house-prices-advanced-regression-techniques | 956K | machine_learning |
| playground-series-s3e1 | 6.2M | machine_learning |
| playground-series-s3e11 | 48M | machine_learning |
| playground-series-s3e13 | 292K | machine_learning |
| playground-series-s3e14 | 2.9M | machine_learning |
| playground-series-s3e15 | 1.7M | machine_learning |
| playground-series-s3e16 | 8.7M | machine_learning |
| playground-series-s3e19 | 13M | time_series |
| playground-series-s3e21 | 672K | machine_learning |
| playground-series-s3e22 | 388K | machine_learning |
| playground-series-s3e24 | 22M | machine_learning |
| playground-series-s3e25 | 2.1M | machine_learning |
| playground-series-s3e26 | 1.4M | machine_learning |
| playground-series-s3e3 | 456K | machine_learning |
| playground-series-s3e5 | 232K | machine_learning |
| playground-series-s3e7 | 3.7M | machine_learning |
| playground-series-s3e9 | 488K | machine_learning |
| playground-series-s4e1 | 21M | machine_learning |
| playground-series-s4e10 | 6.0M | machine_learning |

*Table S4.* Competition Dataset Sizes categorized by difficulty and source

| Competition | Data Size | Domain |
|---|---|---|
| playground-series-s4e11 | 27M | machine_learning |
| playground-series-s4e12 | 318M | machine_learning |
| playground-series-s4e2 | 4.4M | machine_learning |
| playground-series-s4e3 | 5.3M | machine_learning |
| playground-series-s4e4 | 8.1M | machine_learning |
| playground-series-s4e5 | 43M | machine_learning |
| playground-series-s4e6 | 16M | machine_learning |
| playground-series-s4e7 | 1.1G | machine_learning |
| playground-series-s4e8 | 285M | machine_learning |
| playground-series-s4e9 | 46M | machine_learning |
| playground-series-s5e1 | 21M | time_series |
| playground-series-s5e2 | 39M | machine_learning |
| playground-series-s5e3 | 188K | machine_learning |
| playground-series-s5e4 | 91M | machine_learning |
| playground-series-s5e5 | 48M | machine_learning |
| playground-series-s5e6 | 49M | machine_learning |
| playground-series-s5e7 | 1.1M | machine_learning |
| playground-series-s5e8 | 86M | machine_learning |
| titanic | 100K | machine_learning |
| **DSPREDICT-HARD** | | |
| ashrae-energy-prediction | 2.5G | time_series |
| career-con-2019 | 95M | sensor_signal |
| champs-scalar-coupling | 1.6G | chemistry |
| data-science-bowl-2018 | 480M | computer_vision |
| digit-recognizer | 123M | computer_vision |
| elo-merchant-category-recommendation | 2.9G | business |
| gendered-pronoun-resolution | 7.5M | nlp |
| geolifeclef-2024 | 3.3G | geology |
| google-smartphone-decimeter-challenge | 12G | sensor_signal |
| home-credit-default-risk | 2.5G | machine_learning |
| home-data-for-ml-course | 1.2M | machine_learning |
| humpback-whale-identification | 5.7G | computer_vision |
| ieee-fraud-detection | 1.3G | machine_learning |
| imaterialist-challenge-fashion-2018 | 378M | computer_vision |
| imaterialist-challenge-furniture-2018 | 47M | computer_vision |
| inclusive-images-challenge | 16G | computer_vision |
| LANL-Earthquake-Prediction | 9.8G | sensor_signal |
| liverpool-ion-switching | 140M | biology |
| m5-forecasting-accuracy | 430M | time_series |
| m5-forecasting-uncertainty | 492M | time_series |
| march-machine-learning-mania-2023 | 138M | sports |
| march-machine-learning-mania-2025 | 175M | sports |
| mens-machine-learning-competition-2018 | 1.6G | sports |
| mens-machine-learning-competition-2019 | 1.8G | sports |
| mens-march-mania-2022 | 228M | sports |
| microsoft-malware-prediction | 7.9G | machine_learning |
| nlp-getting-started | 1.4M | nlp |
| novozymes-enzyme-stability-prediction | 16M | chemistry |
| open-problems-single-cell-perturbations | 4.3G | bioinformatics |
| otto-recommender-system | 12G | recommender_system |

*Table S4.* Competition Dataset Sizes categorized by difficulty and source

| Competition | Data Size | Domain |
|---|---|---|
| pku-autonomous-driving | 5.9G | computer_vision |
| planttraits2024 | 3.4G | computer_vision |
| predict-ai-model-runtime | 6.9G | machine_learning |
| recruit-restaurant-visitor-forecasting | 136M | time_series |
| rsna-pneumonia-detection-challenge | 3.8G | computer_vision |
| santander-customer-transaction-prediction | 579M | machine_learning |
| santander-value-prediction-challenge | 1.1G | machine_learning |
| siim-acr-pneumothorax-segmentation | 426M | computer_vision |
| spaceship-titanic | 1.2M | machine_learning |
| sp-society-camera-model-identification | 11G | computer_vision |
| stanford-covid-vaccine | 2.6G | bioinformatics |
| statoil-iceberg-classifier-challenge | 1.7G | computer_vision |
| store-sales-time-series-forecasting | 120M | time_series |
| talkingdata-adtracking-fraud-detection | 11G | machine_learning |
| tensorflow-speech-recognition-challenge | 6.9G | audio_speech |
| tgs-salt-identification-challenge | 720M | computer_vision |
| trec-covid-information-retrieval | 13G | nlp |
| understanding_cloud_organization | 6.0G | computer_vision |
| ventilator-pressure-prediction | 667M | sensor_signal |
| vsb-power-line-fault-detection | 12G | sensor_signal |
| web-traffic-time-series-forecasting | 2.3G | time_series |
| womens-machine-learning-competition-2019 | 19M | sports |
| youtube8m-2018 | 1.1G | computer_vision |
| youtube8m-2019 | 534M | computer_vision |

| **MLEBench-Lite** | | |
|---|---|---|
| aerial-cactus-identification | 236M | computer_vision |
| aptos2019-blindness-detection | 18G | computer_vision |
| denoising-dirty-documents | 239M | computer_vision |
| detecting-insults-in-social-commentary | 4.3M | nlp |
| dog-breed-identification | 1.2G | computer_vision |
| dogs-vs-cats-redux-kernels-edition | 2.0G | computer_vision |
| histopathologic-cancer-detection | 13G | computer_vision |
| jigsaw-toxic-comment-classification-challenge | 186M | nlp |
| leaf-classification | 64M | computer_vision |
| mlsp-2013-birds | 1.2G | audio_speech |
| new-york-city-taxi-fare-prediction | 6.9G | machine_learning |
| nomad2018-predict-transparent-conductors | 21M | chemistry |
| plant-pathology-2020-fgvc7 | 1.2G | computer_vision |
| random-acts-of-pizza | 17M | nlp |
| ranzcr-clip-catheter-line-classification | 19G | computer_vision |
| siim-isic-melanoma-classification | 189G | computer_vision |
| spooky-author-identification | 5.1M | nlp |
| tabular-playground-series-dec-2021 | 704M | machine_learning |
| tabular-playground-series-may-2022 | 597M | machine_learning |
| text-normalization-challenge-english-language | 745M | nlp |
| text-normalization-challenge-russian-language | 1.1G | nlp |
| the-icml-2013-whale-challenge-right-whale-redux | 1.6G | computer_vision |

## E.6. Examples of DSPREDICT

**Dataset:** DSPREDICT-HARD
**Domain:**
**Competition:** web-traffic-time-series-forecasting

```
**CHALLENGE NAME: web-traffic-time-series-forecasting**

Challenge description:
# Web Traffic Time Series Forecasting

## Competition Objective
Forecast future traffic to Wikipedia pages. This competition focuses on the problem
of forecasting the future values of multiple time series, as it has always been one
of the most challenging problems in the field. More specifically, we aim the
competition at testing state-of-the-art methods designed by the participants, on the
problem of forecasting future web traffic for approximately 145,000 Wikipedia
articles.

Sequential or temporal observations emerge in many key real-world problems, ranging
from biological data, financial markets, weather forecasting, to audio and video
processing. The field of time series encapsulates many different problems, ranging
from analysis and inference to classification and forecast.

This competition will run as two stages and involves prediction of actual future
events. There will be a training stage during which the leaderboard is based on
historical data, followed by a stage where participants are scored on real future
events.

You have complete freedom in how to produce your forecasts: e.g. use of univariate vs
multi-variate models, use of metadata (article identifier), hierarchical time series
modeling (for different types of traffic), data augmentation (e.g. using Google
Trends data to extend the dataset), anomaly and outlier detection and cleaning,
different strategies for missing value imputation, and many more types of approaches.

We thank Google Inc. and Voleon for sponsorship of this competition, and Oren Anava
and Vitaly Kuznetsov for organizing it.

Kaggle is excited to partner with research groups to push forward the frontier of
machine learning. Research competitions make use of Kaggle's platform and experience,
but are largely organized by the research group's data science team. Any questions or
concerns regarding the competition data, quality, or topic will be addressed by them.

## Evaluation Criteria
Submissions are evaluated on SMAPE between forecasts and actual values. We define
SMAPE = 0 when the actual and predicted values are both 0.

## Submission Requirements
For each article and day combination (see key.csv), you must predict the web traffic.
The file should contain a header and have the following format:

Id,Visits
bf4edcf969af,0
929ed2bf52b9,0
ff29d0f51d5c0,etc.

Due to the large file size and number of rows, submissions may take a few minutes to
score. Thank you for your patience.

## Prizes
$12,000
$8,000
$5,000
```

Top submissions will also have the opportunity to present their work at the NIPS Time
Series Workshop in Long Beach, California, co-located with the top machine learning
conference NIPS 2017. Attending the workshop is not required to participate in the
competition, however only teams that are attending the workshop will be considered to
present their work.

Attendees presenting in person are responsible for all costs associated with travel,
expenses, and fees to attend NIPS 2017.

## Timeline
This competition has a training phase and a future forecasting phase. During the
training phase, participants build models and predict on historical values. During
the future phase, participants will forecast future traffic values.

September 1st, 2017 – Deadline to accept competition rules.
September 1st, 2017 – Team Merger deadline. This is the last day participants may
join or merge teams.
September 1st, 2017 – Final dataset is released.
September 12th 7:59 PM UTC – Final submission deadline.

Competition winners will be revealed after November 13, 2017.

All deadlines are at 11:59 PM UTC on the corresponding day unless otherwise noted.
The competition organizers reserve the right to update the contest timeline if they
deem it necessary.

## Competition Details
- **Competition Host**: Google
- **Competition Type**: Research Prediction Competition
- **Start Date**: July 13, 2017
- **Close Date**: November 15, 2017
- **Total Prize Pool**: $25,000

## Citation
Maggie, Oren Anava, Vitaly Kuznetsov, and Will Cukierski. Web Traffic Time Series
Forecasting. https://kaggle.com/competitions/web-traffic-time-series-forecasting,
2017. Kaggle.

### Dataset Description:

Web Traffic Time Series Forecasting Forecast future traffic to Wikipedia pages
Dataset Description The training dataset consists of approximately 145k time series.
Each of these time series represent a number of daily views of a different Wikipedia
article, starting from July, 1st, 2015 up until December 31st, 2016. The leaderboard
during the training stage is based on traffic from January, 1st, 2017 up until March
1st, 2017. The second stage will use training data up until September 1st, 2017. The
final ranking of the competition will be based on predictions of daily views between
September 13th, 2017 and November 13th, 2017 for each article in the dataset. You
will submit your forecasts for these dates by September 12th. For each time series,
you are provided the name of the article as well as the type of traffic that this
time series represent (all, mobile, desktop, spider). You may use this metadata and
any other publicly available data to make predictions. Unfortunately, the data source
for this dataset does not distinguish between traffic values of zero and missing
values. A missing value may mean the traffic was zero or that the data is not
available for that day. To reduce the submission file size, each page and date
combination has been given a shorter Id. The mapping between page names and the
submission Id is given in the key files. File descriptions Files used for the first
stage will end in '_1'. Files used for the second stage will end in '_2'. Both will
have identical formats. The complete training data for the second stage will be made
available prior to the second stage. train_*.csv- contains traffic data. This a csv
file where each row corresponds to a particular article and each column correspond to
a particular date. Some entries are missing data. The page names contain the
Wikipedia project (e.g. en.wikipedia.org), type of access (e.g. desktop) and type of

```
agent (e.g. spider). In other words, each article name has the following format:
'name_project_access_agent' (e.g. 'AKB48_zh.wikipedia.org_all-access_spider').
key_*.csv- gives the mapping between the page names and the shortened Id column used
for prediction sample_submission_*.csv- a submission file showing the correct format
License Subject to Competition Rules
```

## F. Case Studies

### F.1. Case Studies of Data Analysis Tasks

---

**Failure Case: Qwen3-Coder-480B**          DSGym-Bio (Single-cell biology)

**Keywords.** Jensen-Shannon divergence, gene expression normalization, highly variable gene identification.
**Task.** Calculate the Jensen-Shannon distance between gene expression distributions of cells from 'blood' vs 'lung' tissues. Use only the top 1000 most variable genes and normalize expression values to probabilities. What is the Jensen-Shannon distance rounded to 4 decimal places?
**Pred: 0.1761     GT: 0.0668**
**Trajectory (simplified).**

> **Assistant:** Load `.h5ad` and check tissue labels to locate `blood` and `lung`.
> **User:** `tissue` labels = {spleen, jejunum, lung, mesenteric lymph node, thoracic lymph node}.
> **Assistant:** Since `blood` is missing, I will use **(mesenteric+thoracic) lymph node** as a blood proxy.
> **Assistant:** Select top-1000 HVGs $\rightarrow$ sum counts per gene in each group $\rightarrow$ normalize to probabilities $\rightarrow$ compute JS distance.
> **Assistant:** JS distance = 0.1761.

**Error analysis.** The agent made incorrect assumptions about the blood proxy after finding that no cells are labeled as 'blood' in the metadata, it silently substituted mesenteric+thoracic lymph node cells as a proxy for blood, which changes the underlying distributions and the Jensen-Shannon distance. In contrast, human domain experts chose spleen which is a common blood proxy.
**Error type.** Domain Grounding Error

---

**Failure Case: GPT 5.1**          DSGym-Bio (single-cell biology)

**Keywords.** Differential expression analysis
**Task.** Among the three genes ESR1, FOXA1, and GATA3, which gene shows the highest mean RNA expression in LHS cells compared to LASP cells?
**Pred: GATA3,     GT: ESR1**
**Trajectory (simplified).**

> **Assistant:** Load `.h5ad` and check its metadata stored in 'obs'.
> **User:** Dataset metadata lists `obs` columns including `donor_age`, `author_cell_type`, ..., `LHS_sub`, `LASP_sub`.
> **Assistant:** Can observe `LHS_sub`, `LASP_sub` and assume these are the two cell type gene expression. Define LHS cells by `LHS_sub.startswith("LHS")` and LASP cells by `LASP_sub.startswith("LASP")`; map ESR1/FOXA1/GATA3 via `feature_name`.
> **Assistant:** Compute group means from `adata.X`; observe GATA3 has largest (LHS−LASP) mean difference; predict `GATA3`.

**Error analysis.** The agent inspected `adata.obs` and found the fields `LHS_sub` and `LASP_sub`, then incorrectly assumed they were the primary cell-type annotations for defining LHS and LASP cells. In reality, these columns represent *subtype* labels within the LHS and LASP lineages, not the top-level cell-type assignment. The correct LHS/LASP cell-type identity is provided in `obs["author_cell_type"]`, but the agent did not use this field, leading to an incorrect grouping and downstream mean-expression comparison.
**Error type.** Domain Grounding Error.

---

**Failure Case: GPT 5.1**                                                    DSGym-Bio (Single-cell biology)

**Keywords.** Variance analysis, Cell type composition

**Task.** Among the top 2000 most variable genes, identify the gene with the strongest donor-specific expression pattern while controlling for cell type composition. The donor-specificity score is variance_between_donors / (variance_within_donors + cell_type_effect + 0.01). Which gene has the highest donor-specificity score?

**Pred: ENSG00000078369    GT: ENSG00000135222**

**Trajectory (simplified).**

> **Assistant:** Load the `.h5ad`; use `donor_id`/`Pool` and `cell_type`; compute donor-specificity score via per-gene linear model (`expr ~ Pool + cell_type`).
> **User:** AnnData: 51,367 cells × 35,477 genes; `Pool` has 16 levels; `cell_type` has 8 levels.
> **Assistant:** Treat `Pool` as donor ID; subset valid cells; compute top-2000 most variable genes using sparse moments.
> **User:** Top-2000 selected; first few IDs include `ENSG00000078369`, `ENSG00000171735`, ...
> **Assistant:** Attempt OLS decomposition for each of 2000 genes (dense 51k×2000 + one-hot design + loop) to compute score.
> **User:** `TimeoutError` (computation exceeded limit).
> **Assistant:** Unable to finish scoring; return **first** gene from the variable-gene list: `ENSG00000078369`.

**Error analysis.** The agent never computed the donor-specificity score for the 2000 genes because its OLS-per-gene loop (including dense conversion and per-gene predictions) timed out, and it then guessed by returning the first gene from the variable-gene list. Additionally, its linear-model setup is numerically problematic (full one-hot for both Pool and cell_type without an intercept makes X'X singular), so even without the timeout the variance decomposition could fail or be incorrect. Ground truth answer calculated variance_between_donors, variance_within_donors, and cell type effect (variance of mean expression across cell types) directly, thus bypassed the OLS and didn't run into timeout issues.

**Error type.** Statistical Knowledge Error, Planning Error.

---

**Failure Case: Claude 4 Sonnet**                                            DSGym-Bio (Single-cell biology)

**Keywords.** Pseudobulk differential expression, Statistical testing, Differential expression analysis"

**Task.** For Goblet cells, calculate specificity score: (mean_ileum / sum_of_all_means) x log2_fold_change_vs_others. Filter: mean expression > 1.0 in Ileum, Wilcoxon p-value < 0.01. What are the top 3 gene symbols with highest specificity score?

**Pred: ['REG4', 'REP15', 'CLCA1']    GT: ['RBP2', 'ALDOB', 'APOA1']**

**Trajectory (simplified).**

> **Assistant:** Load Ileum/Colon/Rectum `.h5ad`; locate `Goblet` cells and compute the specificity score.
> **User:** `CellType` includes `Goblet` in all tissues (Ileum: 303; Colon: 765; Rectum: 899).
> **Assistant:** Define *others* as **all non-Goblet cells across all tissues**; run Wilcoxon (Ileum Goblet vs others) and compute log2FC.
> **User:** Gene sets differ across tissues; need intersection.
> **Assistant:** Subset to common genes; compute `mean_ileum_goblet`, `mean_others`, and `sum_of_all_means` as **(ileum mean + colon mean + rectum mean)** over **all cells**.
> **Assistant:** Filter (mean_ileum > 1, $p < 0.01$); score = (mean_ileum/sum_of_all_means) × $\log_2$ FC; top-3 = [REG4, REP15, CLCA1].

**Error analysis.** The agent misinterpreted the contrast and denominator: it compared Ileum Goblet cells against all non-Goblet cells across all tissues, and computed sum_of_all_means from whole-tissue means, whereas the task intends a Goblet-cell tissue-specificity calculation (Ileum Goblet vs Goblet in other tissues, with sum_of_all_means taken over Goblet means across tissues). This population mismatch drives selection of canonical Goblet markers (REG4/CLCA1) instead of Ileum-specific genes expected under the correct definition.

**Error type.** Instruction Following Error, Domain Grounding Error

---

**Failure Case: Qwen3-Coder-480B**                                           DAEval

**Keywords.** Comprehensive Data Preprocessing, Missing Values Handling

**Task.** Perform comprehensive data preprocessing for the dataset by handling missing values in the age and cabin columns. Use the deletion strategy for the missing values in the cabin column and imputation strategy for the missing

values in the age column.

For the deletion strategy in the cabin column, remove any row that has a missing value in the cabin column.For the imputation strategy in the age column, replace the missing values with the median age of all passengers.

Report on the new total number of rows after deletion and the median age used for imputation.

**Pred: @row_count[204], @median_age[28.0]    GT: @row_count[204], @median_age[36.0]**

**Error analysis.** The agent computed the imputation median on the full dataset *before* applying the `Cabin` deletion step. However, the task requires computing the median age *after* deleting rows with missing `Cabin`.

**Error type.** Instruction-following error

---

**Failure Case: Claude 4 Sonnet**                                                                     DAEval

**Keywords.** Correlation Analysis

**Task.** Which variable has the strongest correlation with the happiness scores among countries? Compute the Pearson correlation coefficient between `Happiness Score` and *all other numerical variables*. The variable with the largest $|r|$ (ignoring sign) is the strongest; report whether the correlation is positive or negative.

**Pred: Economy (GDP per Capita), positive    GT: Happiness Rank, negative**

**Error analysis.** The agent explicitly excluded `Happiness Rank` from the candidate set, assuming it is derived from `Happiness Score` and therefore should be removed. However, the task requires correlating `Happiness Score` with *all* numerical columns and `Happiness Score` should not be removed.

**Error type.** Instruction Following error

---

## F.2. Case Studies of Data Prediction Tasks

**Kaggle Case Studies: GPT-5.1 (medium)**                                                           DSPredict-Hard

**Competition Name.** pku-autonomous-driving

**Task.** Can you predict vehicle angle in different settings? This challenge involves estimating the 6-Degrees-of-Freedom (6-DOF) absolute pose—comprising 3D translation $(x, y, z)$ and rotation (pitch, yaw, roll)—for all unmasked vehicles in monocular street-level images. The dataset provides industry-grade 3D CAD car models and camera intrinsic parameters to facilitate geometric mapping from 2D image coordinates to 3D world space. Success requires accurate spatial reasoning to predict orientation relative to the camera and precise distance estimation, evaluated via mean Average Precision (mAP) across tight translational and rotational thresholds.

**Trajectory (simplified).**

**Assistant:** Load and parse `train.csv` to calculate global median pose values for pitch, yaw, roll, x, y, and z.
```
all_yaws = [obj[1] for obj in all_objs]
y_med = np.median(all_yaws) # median of all pose components
```

**Env:** `train.csv` loaded (4,262 images). Median car count: 11. Median $z$-coordinate: 44.89m.

**Assistant:** Attempt to use `cv2` to segment car instances from binary masks to improve car count estimation.
```
import cv2
num_labels, labels = cv2.connectedComponents(bin_mask)
```

**Env:** `ModuleNotFoundError:  No module named 'cv2'.`

**Assistant:** Check if total mask foreground area correlates with the number of cars.
```
correlation = subset[["num_cars", "mask_fg_area"]].corr()
```

**Env:** Correlation is $-0.0056$; mask area is not a valid feature for car count.

**Assistant:** Try counting unique non-zero pixel values in masks as a proxy for the number of car instances.
```
n_instances = len(np.unique(mask)[np.unique(mask) > 0])
```

**Env:** Correlation is 0.0336; unique mask values do not match ground truth counts.

**Assistant:** Final Fallback: Generate a submission predicting exactly 11 cars per test image using the global medians.
```
pred = " ".join([f"{p_med} {y_med} {r_med} {x_med} {y_med} {z_med} 1.0"] * 11)
submission_df.to_csv("/submission/submission.csv", index=False)
```

**Env:** `submission.csv` successfully saved with 2,021 rows.

**Valid Submission: Yes    Medal: None    Above Median: No    Percentile: 4.57**

**Analysis.** Predicted 11 identical cars for every test image using global median pose values and fixed confidence, ignoring image cues, masks, or per-image car counts. This case illustrates simplicity bias: the agent prioritizes "Valid Submission" over "Model Performance." When its primary tool (`cv2`) failed and simple correlation tests proved unfruitful, the agent opted for a statistically trivial median-based fallback rather than pursuing nuanced 3D geometry or image-based feature engineering. This minimizes trajectory length at the expense of competitive accuracy.

**Type: Environment Access Restrictions**

---

**Kaggle Case Studies: GPT-5.1 (medium)**                                                   DSPredict-Hard

**Competition Name.** web-traffic-time-series-forecasting

**Description.** This challenge involves forecasting daily traffic for  145,000 Wikipedia articles. The task is characterized by high-cardinality, non-stationary data, and a high volume of missing values. Performance is measured by the Symmetric Mean Absolute Percentage Error (SMAPE), which penalizes over- and under-predictions symmetrically on a percentage basis.

**Trajectory (simplified).**

> **Assistant:** Initialize environment and optimize memory by downcasting data to `float32` and filling NaNs with 0.
> ```
> train2[date_cols] = train2[date_cols].fillna(0.0).astype("float32")
> ```
>
> **Env:** Memory usage reduced; 145,063 time series loaded across 803 dates.
>
> **Assistant:** Establish a pseudo-forecast validation window (62 days) to test multiple baseline heuristics.
> ```
> def median_last_k(hist_array, k):
>     return np.repeat(np.median(hist_array[:, -k:], axis=1)[:, None], 62, axis=1)
> ```
>
> **Env:** Baseline SMAPE: Naive Last (0.462), Mean 30d (0.461), Weekly Seasonal (0.416).
>
> **Assistant:** Identify 30-day median as the optimal heuristic (SMAPE: 0.404) and proceed with page-to-key mapping.
> ```
> key2["base_page"] = key2["Page"].str.slice(0, -11) # Strip _YYYY-MM-DD
> merged = key2.merge(page_median, how="left", on="base_page")
> ```
>
> **Env:** Successfully mapped all 8,993,906 forecast IDs to historical medians.
>
> **Assistant:** Generate final submission by rounding medians to non-negative integers.
> ```
> merged["Visits"] = np.round(merged["median_last30"]).clip(lower=0)
> ```
>
> **Env:** `submission.csv` generated (8,993,906 rows). Validation SMAPE: 0.404.

**Valid Submission: Yes    Medal: None    Above Median: Yes    Percentile: 51.33**

**Analysis.** Rather than defaulting to compute-heavy recurrent architectures (e.g., LSTM/GRU) typically associated with this competition, the agent uses a systematic validation framework to prove that a simple 30-day median is more robust than seasonal or naive averages. This selection prioritizes safety in the face of noisy, missing data, ensuring a valid submission that outperforms more "sophisticated" but less stable deep learning models.

**Type: Low-Effort Heuristics**

## F.3. Examples of Shortcut Solvable Tasks

In this section, we present representative examples of *shortcut-solvable* tasks (tasks that can be answered correctly without data files). For each case, we analyze the agent trajectory and identify the underlying shortcut mechanism. These examples illustrate how correctness in existing benchmarks may arise from prompt-level reasoning, domain priors, or contamination, rather than genuine data interaction.

**Example of Shortcut Solvability via Domain Priors**

**Task.** A multiple-choice question from QRDATA asks: *"Is the `smoke` variable numerical or categorical?"* The task is file-grounded and provides a dataset (`smoking.csv`), but the agent is evaluated without access to the data files.
**Trajectory (simplified).** The agent directly produces a final answer without attempting any data access or code execution:

> **Assistant:** Reason about the semantic meaning of the variable name `smoke` and apply general statistical conventions. Infer that `smoke` typically encodes smoking status (e.g., smoker vs. non-smoker or discrete frequency categories), and therefore corresponds to a categorical variable. <answer>categorical</answer>

**Analysis.** This is a *domain-prior shortcut*. Correctness arises from widely shared semantic conventions and textbook-level priors about variable naming.

---

**Example of Shortcut Solvability via Contamination**

**Task.** A multiple-choice question from QRDATA asks: *"In which group did a higher percentage of patients become pain free 24 hours after receiving acupuncture?"* The task is file-grounded and provides a dataset (`migraine.csv`), but the agent is evaluated without access to the data files.
**Trajectory (simplified).**

> **Assistant:** Identify the question as a canonical randomized controlled trial example from an introductory statistics textbook. Try to load available data although not allowed to see it.
> ```python
> print("Current directory contents:")
> print(os.listdir('.'))
> for root, dirs, files in os.walk('.'):
>     for file in files:
>         if file.endswith(('.csv', '.txt', '.xlsx', '.json')):
>             print(f\"Found data file: {os.path.join(root, file)}\")
> ```

> **Env:** Output only code files and no data files.

> **Assistant:** Recall the reported outcome that a higher proportion of patients in the treatment group became pain free after 24 hours.
> ```python
> # This is a well-known case study about acupuncture treatment for chronic pain
> # Treatment group (actual acupuncture): 10 out of 43 patients became pain-free
> # Control group (sham acupuncture): 2 out of 46 patients became pain-free
> treatment_pain_free = 10
> treatment_total = 43
> control_pain_free = 2
> control_total = 46
> print("Treatment Group:")
> print(f"Pain-free patients: {treatment_pain_free}")
> print(f"Total patients: {treatment_total}")
> print(\"\nControl Group:")
> print(f"Pain-free patients: {control_pain_free}")
> print(f"Total patients: {control_total}")
> ```

> **Env:** Output the variables defined in the code.

> **Assistant:** Calculate the percentage of patients who became pain-free in each group and compare them to determine which group had a higher percentage.
> ```python
> treatment_percentage = (treatment_pain_free / treatment_total) * 100
> control_percentage = (control_pain_free / control_total) * 100
> print(f"Treatment Group percentage pain-free: {treatment_percentage:.2f}%")
> print(f"Control Group percentage pain-free: {control_percentage:.2f}%")
> ```

> **Env:** Treatment Group percentage pain-free: 23.26% Control Group percentage pain-free: 4.35%

> **Assistant:** Analyze the outputs and come to the final answer. <answer>treatment group</answer>

**Analysis.** This example exhibits a *contamination-driven shortcut*. Although the agent's reasoning mirrors a valid

statistical comparison, the numerical quantities are not derived from data interaction but are effectively recalled from memorization.

## G. Experiment Details

### G.1. Details of Evaluation

**Models.** We benchmark the following models through DSGym: **GPT-5.1** (`gpt-5.1-2025-11-13`), **GPT-5** (`gpt-5-2025-08-07`) (OpenAI, 2025a), **GPT-4o** (`gpt-4o-2024-08-06`) (OpenAI, 2024), **Claude Sonnet 4.5** (`claude-sonnet-4-5-20250929`) (Anthropic, 2025b), **Claude Sonnet 4** (`claude-sonnet-4-20250514`) (Anthropic, 2025a), **Qwen3-235B-Instruct** (`Qwen3-235B-A22B-Instruct-2507-tput`) (Yang et al., 2025a), **QWEN3-CODER 480B** (`Qwen3-Coder-480B-A35B-Instruct-FP8`) (Yang et al., 2025a), **Kimi K2 Instruct** (`Kimi-K2-Instruct-0905`) (Team et al., 2025), **GPT-OSS-120B** (`gpt-oss-120b`) (OpenAI, 2025b), **Deepseek-v3.1** (`DeepSeek-V3.1`) (DeepSeek-AI, 2025). We also include **Qwen2.5-7B-Instruct** (Qwen et al., 2025) and **Qwen3-4B-Instruct** (Yang et al., 2025a) as open-source small models and **Datamind-7B** (Qiao et al., 2025) as a baseline for **Qwen3-4B-Instruct-DSGym-SFT-2K** and **Qwen2.5-Coder-7B-DSGym-SFT-2K**. For Datamind-7B, we directly utilize the checkpoint and system prompt provided in the original paper. For all the other models, we utilize the same system prompt as shown in Appendix. H.

**Hyperparameters.** We set temperature=0 for all models during evaluation. For GPT-5, the reasoning effort is set to `medium` as default. For GPT-5.1, we vary different reasoning efforts from `none` to `medium` and `hard`.

**Computational Cost.** For DSBIO, we impose a per-turn execution limit of 20 minutes, matching expert review: all reference solutions complete within this budget. For general analysis tasks, we use a 3-minute per-turn limit. Empirically, when the code is reasonable, agents will not time out under these settings. We employ a max turn of 30 for all analysis tasks. For DSPREDICT, each agent is subject to a strict total runtime budget of 10 hours. Within each run, individual actions are further constrained to 2 hours in Table 4 and 4.5 hours in Table S1. Agents are provisioned with a single A100 GPU and are explicitly informed of its availability. Actual session runtimes vary substantially, as agents adopt different solution strategies. In practice, the dominant bottleneck is code execution—primarily model training—rather than LLM inference. Consequently, session durations range from a few minutes to the 10-hour limit. Across models, the average runtime typically falls between 2 and 5.5 hours per competition. When evaluated on a node equipped with eight A100 GPUs, completing DSPREDICT-HARD requires approximately 8–12 hours, while DSPREDICT-EASY can be evaluated in 2–5 hours.

### G.2. Details of Training

We integrate LlamaFactory (Zheng et al., 2024) into DSGym for SFT training. Our learning rate is 2e-5 with a warmup ratio of 0.1 and a cosine decay schedule. The detailed hyperparameters employed are presented in Tab.S5.

*Table S5.* Detailed hyperparameters used in our paper.

| Stage | Hyperparameter | Value |
|---|---|---|
| SFT | learning rate | 2e-5 |
| | lr scheduler type | cosine |
| | warmup ratio | 0.1 |
| | batch size | 8 |
| | training epoch | 6 |
| | gradient accumulation steps | 16 |
| | neftune noise alpha | 10 |
| Inference | temperature | 0 |
| | top p | 1 |

### G.3. Details of DSPREDICT Evaluation Metrics

Each agent is given access to a single A100 GPU. Agent is not required to use it but is informed of its existence.

**Public and Private Leaderboard Evaluation** To ensure rigorous and consistent assessment, DSPREDICT results are evaluated using the official Kaggle scoring logic. Standard Kaggle competitions utilize a dual-leaderboard system: a Public leaderboard for real-time feedback and a Private leaderboard—calculated on a withheld data split—to determine final standings and mitigate overfitting. We utilize the Kaggle API to programmatically submit agent-generated solutions to Kaggle's servers, retrieving authentic scores and percentile standings.

In our primary analysis (Table 4), we report exclusively on Private Leaderboard scores to provide a more conservative and robust measure of generalization. It should be noted that certain archived competitions no longer support active submissions or have closed their private leaderboard visibility. Consequently, we report results for the subsets where private scores remain obtainable: 42 of 54 competitions for DSPREDICT-HARD and 36 of 38 for DSPREDCIT-EASY. To ensure reproducibility, we report results using a fixed offline snapshot of the leaderboard, since the online leaderboard may change over time. All results are tied to a snapshot with a cutoff date of 01/29/2026. If any competition doesn't accept new submission anymore, we will remove it form DSPREDICT and potentially add new competitions.

**Reproducibility and Data Licensing Compliance** For DSPREDICT , our evaluation is conducted in a Dockerized testbed consisting of 8 executor containers and 1 manager container. Each executor is allocated 8 CPU cores and 24 GB RAM. All experiments are run using a fixed Docker image built from `pytorch/pytorch:2.8.0-cuda12.6-cudnn9-runtime @sha256:dab81780fd94483b67b4b5679cc0024939b08e48540d39476d284cb29002ed69`, which specifies CUDA 12.6 and cuDNN 9. Dependencies are installed at build time via a pinned requirements.kaggle.txt file (including libraries such as NumPy and XGBoost), ensuring a consistent Python environment across runs. The container is configured to run under a non-root user, with a fixed working directory (/app) to standardize execution behavior.

We recognize the importance of licensing integrity when using Kaggle-sourced data. To ensure full compliance with Kaggle's Terms of Service, DSPREDICT adopts a "Federated Download" model rather than direct redistribution of raw data. We do not host or mirror restricted Kaggle datasets. Instead, we provide standardized Task Specification Files and automated download-and-preprocess scripts. Users are required to provide their own Kaggle API (easy to obtain) credentials to fetch the data directly from Kaggle's servers. This ensures that every user has explicitly agreed to the specific dataset's Terms of Use and Competition Rules prior to access. This approach mirrors the standard practices of established benchmarks like MLE-bench, ensuring the framework is legally robust for long-term academic and commercial use. To support reproducibility in the presence of a dynamic online leaderboard, we release a fixed offline snapshot of leaderboard. Although model performance may evolve in future online evaluations, all results reported in this paper are anchored to this offline snapshot, defined by a time cutoff of 01/29/2026.

**Metrics** To assess the performance of models in DSPREDICT, we require additional metrics below:

1. **Valid Submission:** A submission to a competition is considered valid if and only if a correctly formatted `submission.csv` file is generated. To be valid, the file must exist, and both the number of items and the column headers must strictly match the competition requirements.
2. **Above Median:** Each competition is associated with a leaderboard. An agent's run is considered "Above Median" if the final score of the submission exceeds the median score of the leaderboard.
3. **Percentile:** This metric represents the agent's relative standing on the leaderboard. For example, a percentile of 30 indicates that the agent's score outperformed 30% of all other submissions.
4. **Medal:** Kaggle awards Bronze, Silver, and Gold medals based on leaderboard performance. We follow MLEBench (Chan et al., 2024) to determine medal acquisition. The thresholds for Bronze, Silver, and Gold vary based on the number of teams in the competition. Table S6 illustrates the logic for awarding medals.

## H. Prompts

In this section, we provide the prompts we use for evaluation and training.

*Table S6.* Kaggle Medal thresholds based on the number of participating teams.

| | 0-99 Teams | 100-249 Teams | 250-999 Teams | 1000+ Teams |
|---|---|---|---|---|
| **Bronze** | Top 40% | Top 40% | Top 100 | Top 10% |
| **Silver** | Top 20% | Top 20% | Top 50 | Top 5% |
| **Gold** | Top 10% | Top 10 | Top 10 + 0.2%* | Top 10 + 0.2%* |

## H.1. System Prompt

**System Prompt for Data Prediction Tasks**

You are an expert data scientist and machine learning engineer who tackles modeling and machine learning challenges through systematic thinking, investigation and rigorous evaluation. For each task, you will receive a challenge description along with file paths to the training and test data.

Your goal is to:

1. Understand the problem — interpret the competition objective, data format, and evaluation metric.
2. Explore and preprocess the data — load the datasets, perform data cleaning, feature engineering, and exploratory analysis where helpful.
3. Decompose the question and perform planning - break down the task into smaller steps and perform each step systematically. Change your plan if needed.
4. Train and validate models — build competitive ML models with proper validation strategies to avoid overfitting.
5. Generate predictions — apply the trained model to the test set and produce a submission.csv file in the required format.
6. Explain reasoning — clearly communicate assumptions, methodology, and trade-offs at each step.

Important Rules:

- Do not use plotting libraries (you cannot view plots). Use text-based summaries and statistics instead.
- Try different approaches or perform deeper reasoning when your model is not performing well.
- You can split the training data into training and validation set to tune your model until you are satisfied with the performance.
- Code execution is continuous - variables and data loaded in previous steps remain available for subsequent steps. Do not need to reload the same dataset or variables.
- Your code can only do one step at a time even when multiple steps are planned. Perform the next step based on the previous step's results.
- After you produce the submission.csv, you must check the format of this file according to the competition requirements.
- When you decide to finish the task after producing the submission.csv, You must provide your concise summary in the format: `<answer>your final summary</answer>`

You MUST use the following format for your response. Each step must follow this exact structure:

```
<reasoning>
Write clear reasoning about what you plan to do next and why. Be specific about your analytical approach.
<reasoning>

Write executable Python code here. Each code block should do ONE specific task.
Code must be complete and runnable. Include all necessary imports.

<information>
The output/results from your Python code will appear here. This section is read-only - you cannot write here.
</information>
```

Repeat these blocks for each analysis step. When you reach your conclusion, you should follow this structure:

```
<reasoning>
```
Write clear reasoning about how you came up with your final answer.
```
</reasoning>
<answer>
```
Write a concise summary/answer here. Do not include any other text or unnecessary information.
```
</answer>
```

## System Prompt for Data Analysis Tasks

You are an expert data scientist, statistical analyst and machine learning engineer who tackles analytical or machine learning challenges through systematic thinking and investigation. For each task, you will receive a question along with file paths to the relevant data and background information.

Your goal is to:

1. Understand the problem — interpret the question, data format, and expected output format.
2. Explore and preprocess the data — load the datasets, perform data cleaning, feature engineering, and exploratory analysis where helpful.
3. Decompose the question and perform planning - break down the question into smaller steps and perform each step systematically. Change your plan if needed.
4. Analyze the data — build appropriate statistical models, causal models, machine learning models, or other analyses to answer the research question.
5. Generate final answer — provide a clear, specific answer to the question based on your analysis and the requirements.
6. Explain reasoning — clearly communicate assumptions, methodology, and trade-offs at each step.

Important Rules:

- Do not use plotting libraries (you cannot view plots). Use text-based summaries and statistics instead.
- Your final answer should be specific and directly address the question.
- For numerical answers, provide the exact value requested (rounded as specified if mentioned).
- Only produce the final answer when you have enough evidence and validation to support your approach.
- Try different approaches or perform deeper reasoning when you are uncertain about the answer.
- Code execution is continuous - variables and data loaded in previous steps remain available for subsequent steps. Do not need to reload the same dataset or variables.
- Your code can only do one step at a time even when multiple steps are planned. Perform the next step based on the previous step's results.
- When calculation is needed, you are encouraged to use python code instead of calculating by yourself.
- When you decide to finish the task, you must provide your final answer in the format: `<answer>your final answer</answer>`

You MUST use the following format for your response. Each step must follow this exact structure:

```
<reasoning>
```
Write clear reasoning about what you plan to do next and why. Be specific about your analytical approach.
```
</reasoning>

```
Write executable Python code here. Each code block should do ONE specific task.
Code must be complete and runnable. Include all necessary imports.
```

<information>
```
The output/results from your Python code will appear here. This section is read-only - you cannot write here.

```
</information>
```

Repeat these blocks for each analysis step. When you reach your conclusion, you should follow this structure:

```
<reasoning>
```
Write clear reasoning about how you came up with your final answer.
```
</reasoning>
<answer>
```
Write your final answer here according to the requirements of the question. Do not include any other text or unnecessary information.
```
</answer>
```

## H.2. User Prompt

**User Prompt Abstraction**

TASK: `<task description>`

DATASET INFORMATION:

`<dataset information>`

DATASET LOCATIONS:

`<docker_data_path>`

INSTRUCTIONS:

`<instructions>`

---

**User Prompt for Data Prediction Tasks**

TASK: Tackle the given Kaggle challenge by training ML models on training data to provide a final submission.csv.

COMPETITION NAME: `<challenge_name>`

COMPETITION INTRODUCTION:
`<introduction of this competition>`

DATASET INFORMATION:
`<dataset information>`

DATASET LOCATIONS (this is the path of the directory):
`<docker_data_path>`

INSTRUCTIONS:
1. Load and explore the **training** and **test** datasets using Python (use the dataset folder location provided).
2. Perform **data preprocessing** (handling missing values, encoding, scaling, feature engineering) and **exploratory analysis** to understand distributions, correlations, and relationships between variables.
3. Where simple preprocessing and baseline models are insufficient, attempt more advanced approaches such as:
   • Model selection (e.g., tree-based models, linear models, neural networks)
   • Cross-validation and hyperparameter tuning
   • Dimensionality reduction, feature selection, or ensembling
   • Robustness checks or combining datasets if useful
4. Use the training data to build a model, evaluate it with proper validation, and then generate **predictions for the test data**.

5. Do one step at a time. Explore and validate thoroughly before moving on to model training and submission.

6. When doing exploration and data analysis, print the results in a clear and concise way.

7. Do not use plotting libraries (assume you cannot view plots). Use text-based summaries and statistics instead.

8. When workflow tags or competition-specific guidelines are provided, you should follow them closely.

9. Only produce the **final submission and answer** when you have enough evidence and validation to support your approach.

10. When you finished training the best model, you should generate the final submission:

    (a) Use the best model to generate predictions for the test data located at the path shown above.

    (b) Save predictions in the required **'submission.csv' format** for the competition at /submission/submission.csv.

    (c) Provide a concise summary of your approach in the format: ¡answer¿your final summary¡/answer¿

---

**User Prompt for Data Analysis Tasks**

TASK: `<task description for the dataset>`

QUESTION: `<question statement><answer guidelines (if any)>`

`<question information>`

DATASET INFORMATION:
`<dataset information>`

DATASET LOCATIONS (this is the path of the directory):
`<docker_data_path>`

INSTRUCTIONS:

1. Load and explore the provided datasets using Python.

2. Consider useful python libraries such as pandas, numpy, scipy, scikit-learn, statsmodels, dowhy, econml, causalml, linearmodels, networkx, etc.

3. Apply appropriate statistical methods or analysis techniques to answer the research question.

4. Your final answer should be specific and directly address the question. Do not include any other text. e.g., <answer>0.23</answer>

