# OpenReview forum: "DSGym: A Standardized and Holistic Framework for Evaluating and Training Data Science Agents"
_ICML.cc/2026/Conference — ICML 2026 regular_

### Official Review · Reviewer_ZPVH · 2026-03-11

**Soundness:** 3
**Presentation:** 3
**Significance:** 2
**Originality:** 3
**Overall Recommendation:** 4
**Confidence:** 5

**Summary:**

This paper introduces DSGym, a framework for evaluating and training LLM-based data science agents. The core contributions are: (1) a standardized execution environment using containerized Jupyter kernels, (2) a curated task suite (DSGym-Tasks) with analysis and filtering of shortcut-solvable tasks, (3) two new task datasets (DSBio and DSPredict), and (4) a trajectory generation pipeline for synthesizing agent training data. The authors also benchmark a range of frontier models and identify persistent failure modes.

**Compliance With Llm Reviewing Policy:**

Affirmed.

**Key Questions For Authors:**

- How can DSGym remain a reliable testbed as LLM capabilities continue to improve? In particular, what process will be used to periodically re-evaluate or reapply the shortcut filtering to ensure that tasks do not become solvable via new shortcut strategies as models advance?

- What empirical evidence demonstrates that DSGym tasks require nontrivial data reasoning rather than superficial code execution (e.g., dataset inspection, simple statistics) or generic analysis pipelines?

**Limitations:**

- The primary contribution of the paper is the construction of a benchmark framework and task suite rather than a new learning algorithm. While this is valuable, the paper may fit more naturally in a dataset/benchmark-oriented track or venue given the nature of the contributions.

- The “without data access” bars in Figure 3 vary significantly across models and benchmarks. Given the concern raised above, the paper should report a full per-model breakdown of shortcut solvability rather than only aggregate statistics, and discuss whether large variation across models could indicate contamination effects rather than purely prompt-level solvability.

- Minor: please name the 4B model in the Abstract for more clarity.

**Strengths And Weaknesses:**

Strengths:
- Exposing the limitations of existing benchmarks for data-reliant tasks is a useful contribution.

- The shortcut solvability finding that a substantial fraction of tasks in existing file-grounded benchmarks can be solved without accessing data files is important and often overlooked in prior evaluation work (Figure 3 and qualitative examples in Appendix F.3).

- Benchmarking 10+ models (both proprietary and open-weight) across analysis, scientific, and prediction tasks provides a useful comparative evaluation of current agent capabilities.

- Demonstrating that a 4B model fine-tuned on 2,000 execution-verified synthetic examples approaches the performance of frontier models on several analysis tasks is a strong proof-of-concept that validates the framework's training utility.


Weaknesses:
- The authors use five models (Claude-Sonnet-4, Qwen3-Coder, DeepSeek-V3.1, GPT-4o, and Kimi-K2-Instruct), which is a reasonable design choice that partially mitigates concerns about correlated contamination. However, the choice of the majority-vote threshold (≥3/5) is not clearly justified. Tasks solvable by two models are retained while those solvable by three are filtered out, but no principled argument is provided for why this boundary is appropriate. A more principled alternative would require tasks to be unsolvable by all evaluated models without data access, or would provide an empirical justification for the chosen threshold.

- The filtered benchmark may reintroduce shortcuts as models improve. A task that no current model can shortcut might be trivially solvable by a future, better-trained model. The paper frames DSGym as a live testbed but does not discuss how or when the shortcut filter should be re-applied, which is a notable gap.

- The scale of DSBio is limited. Ninety tasks across seven papers (Table S3) is a relatively small benchmark. Performance estimates on such a small benchmark will likely have high variance, and the heavy concentration in single-cell biology (55/90 tasks) limits claims about broader bioinformatics capability. Spatial transcriptomics (13/90) and genetics (21/90) are underrepresented.

- All models are evaluated using the same default CodeAct agent at temperature 0. It is therefore unclear how much of the performance differences are attributable to the underlying LLM versus the agent scaffold. While the framework advertises modularity for agent architectures, this is not explored experimentally.

---

> ### Author Rebuttal · Authors · 2026-03-31
>
> We thank the reviewer for the constructive feedback. We provide (1) sensitivity analysis and human validation for shortcut filtering and (2) results using alternative agent architecture in https://anonymous.4open.science/r/ICML2026_rebuttal_DSGym-2847/rebuttal_tables.pdf.
>
> > W1: unsolvable by all models/Threshold empirical justification
>
> The $\ge 3/5$ threshold is an operational criterion backed by sensitivity analysis and human validation in Tables 1 and 2 of the anonymous supplementary materials. A $\ge 1/5$ threshold is too strict (discarding tasks due to single-model lucky guesses), while $5/5$ is too loose; $3/5$ perfectly balances this trade-off. Our human study confirms its reliability, showing 100% precision (all 20 filtered tasks confirmed as shortcut-solvable) and 90.9% recall (20/22 shortcut-solvable tasks correctly identified).
>
> > W2 & Q1: Reintroducing shortcuts as LLMs improve
>
> DSGym is intended as a live testbed rather than a one-time static release. We have established a periodic maintenance protocol: shortcut filtering will be periodically rerun with frontier models, tasks that become newly shortcut-solvable will be retired from the active set, and refreshed releases will be versioned so prior results remain reproducible. Additionally, DSGym's unified task abstraction and common agent interface allow new tasks to be seamlessly added as model capabilities evolve. Importantly, DSPredict tasks are continuous optimization problems and therefore less susceptible to saturation, providing a persistent signal even as models improve.
>
> > W3: scale of DSBio is limited.
>
> DSBio is intended as an initial high-quality scientific benchmark, not a comprehensive biology suite. Our goal is to build an expert-curated, domain-grounded benchmark in a setting that is difficult to construct automatically, and the current concentration in single-cell biology reflects the first area where we established a reliable expert-centered pipeline. We believe this is a valuable trade-off for probing whether models can solve realistic, complex scientific analysis problems. We also intentionally prioritize difficult, reasoning-intensive tasks over routine analyses that often follow standardized SOP-like workflows. As a result, DSBio is smaller, but higher in difficulty and construction quality. A central contribution is our scalable expert-curation workflow, which can be extended to additional papers, subdomains, and scientific areas over time.
>
> > W4: unclear how much of the performance differences are attributable to LLM versus the scaffold. modularity for agent architectures not explored.
>
> We **used a simple shared scaffold (CodeAct) to ensure performance differences reflect underlying LLM capabilities**, not architecture optimizations. To demonstrate DSGym's modularity, we integrated the advanced AIDE scaffold. Due to computational constraints, we present partial results in Table 5 in the rebuttal PDF (AIDE Results on DSPredict-Hard).
> These results provide an initial exploration of scaffold modularity within DSGym.
> Under our current setup, replacing the CodeAct scaffold with AIDE does not consistently improve outcomes, especially on DSPredict, suggesting that scaffold complexity alone does not explain the observed performance differences.
>
> > Q2: DSGym tasks require nontrivial data reasoning?
>
> We here take examples from DSBio and DSPredict as evidence:
> - Tasks in DSBio require genuinely domain-grounded, multi-step reasoning as provided in Appendix E.3 and F.1. For example, endothelial-cell co-expression task (App. E.3) requires modality-specific choices (selecting the top 500 variable genes, hierarchical clustering, dendrogram cutting), which are impossible to solve via generic data profiling.
> - DSPredict tasks are drawn from Kaggle competitions, which are operationally continuous optimization problems over held-out evaluation metrics rather than simply producing runnable code. Case study (App F.2) shows that, basic preprocessing and heuristics yield low percentiles (51st percentile). By contrast, the 1st-place solution optimizes differentiable surrogates using engineered features and recurrent architectures. Strong DSGym performance requires substantive data reasoning and modeling decisions.
>
> > L1: Fit to ICML
>
> Thanks for the suggestion. ICML routinely publishes environment/benchmark track literature (e.g., PaperBench [ICML'25], SWE-Gym [ICML'25]).
>
> > L2: per-model shortcut breakdown & contamination
>
> Thanks for the suggestion. Figure 3 already provides the no-data breakdown for five models; we will expand this to all evaluated models in the appendix. Besides, large variations across models do indicate contamination. But we'd like to clarify that **contamination is one possible source of shortcut solvability**. We've provided examples of shortcut-solvable tasks (e.g., contamination, domain priors) in Appendix F.3.
>
> > L3: name the 4B model
>
> Thanks for the suggestion. We will name it explicitly for clarity.

---

> > ### Author Rebuttal · Reviewer_ZPVH · 2026-04-01
> >
> > Thanks for the rebuttal. The authors mostly addressed the addressable concerns. However, some aspects cannot be addressed in the rebuttal (as the author also mentioned) and are inherent to the work, such as the need for periodic maintenance and the limited scale. My current score is reflective of those limitations.

---

> > > ### Author Response · Authors · 2026-04-07
> > >
> > > We thank the reviewer for the thoughtful feedback and for recognizing the strengths of our work. We agree that periodic maintenance is a limitation of a live benchmark; however, static benchmarks are inherently prone to contamination and shortcut exploitation as model capabilities improve. In this sense, a live, evolving testbed is a principled way to maintain validity over time, and similar periodic maintenance is already common practice in many widely used benchmarks. We greatly appreciate the reviewer’s time and feedback, which help improve the paper!

---

### Official Review · Reviewer_U6JR · 2026-03-12

**Soundness:** 2
**Presentation:** 3
**Significance:** 2
**Originality:** 2
**Overall Recommendation:** 3
**Confidence:** 3

**Summary:**

DSGYM is a standardized, extensible framework designed to evaluate and advance data science agents in reproducible, stateful execution environments. It addresses limitations like fragmented interfaces, shortcut solvability, narrow domain coverage, and lack of rigorous data grounding. The framework features a modular architecture and a curated task suite, which unifies and refines existing benchmarks via quality and shortcut filtering, and expands coverage with two novel suites. Beyond evaluation, DSGYM enables agent training through execution-verified synthetic data generation.

**Compliance With Llm Reviewing Policy:**

Affirmed.

**Final Justification:**

Thank you for your response, and I will maintain my rating.

**Key Questions For Authors:**

refer to the weakness

**Limitations:**

yes

**Strengths And Weaknesses:**

**Strengths**

The paper highlights the shortcut issue in existing data science benchmarks and provides a well-engineered and extensible unified evaluation framework.

**Weaknesses**

1. Limited novelty.


2. The SFT training experiments are relatively limited, making it difficult to draw strong conclusions.

3. The threshold used in Shortcut Filtering is not analyzed for sensitivity, leaving its robustness unclear.

4. Although the Related Work discusses specialized agent architectures such as AIDE, the experiments do not include comparisons with these systems.

---

> ### Author Rebuttal · Authors · 2026-03-31
>
> We thank the reviewer for the feedback. Here we address the raised concerns one by one. We provide additional SFT experiments and sensitivity analysis in https://anonymous.4open.science/r/ICML2026_rebuttal_DSGym-2847/rebuttal_tables.pdf.
>
> > W1: Limited novelty
>
> DSGym advances existing literature by uniquely unifying heterogeneous tasks behind a standardized API, introducing protocol to filter prompt-only shortcuts, and demonstrating that the same execution environment can successfully synthesize training data.
> - The standardization of DSGym is intended to reduce fragmentation across prior benchmarks and to provide reusable infrastructure for both benchmarking and training.
> - We view shortcut filtering as a methodological contribution to benchmark construction. Prior file-grounded tasks often implicitly assume that file access guarantees data dependence. In contrast, **DSGym explicitly shows that a substantial fraction of previous file-grounded tasks can be solved without data**, motivating a new benchmark construction step that removes frequently shortcut-solvable tasks.
> - **DSGym expands the scope through new suites**: expert-derived bioinformatics (DSBio) and real-world predictive modeling (DSPredict).
> - Beyond evaluation, **the same environment supports execution-grounded synthetic data generation** for post-training, illustrating that DSGym serves as a reusable framework for both assessment and agent improvement.
>
> > W2: The SFT training experiments are relatively limited, making it difficult to draw strong conclusions.
>
> To strengthen this section, we provide additional ablation experiments (Table 3):
> - **w/o execution validation/judge criteria/diversity filter**: accuracy all drops without each component on QRData and DABStep. w/o execution causes the largest drop.
> - **Alternative backbone**: Applying DSGym-SFT-2k to Qwen2.5-7B yields similarly strong relative gains (35.04→55.38 on QRData).
> - **Eval on DSPredict** (Table 4): Models trained on synthesized analysis data do not transfer to DSPredict-Hard, but it's expected. Our synthesized data targets analysis workflows, which are qualitatively different from DSPredict tasks. DSGym uniquely supports future work to synthesize task-specific data for DSPredict.
>
> > W3: The threshold used in Shortcut Filtering is not analyzed for sensitivity
>
> To validate the shortcut filtering and $\ge 3/5$ threshold, we provide a sensitivity analysis and human validation study in Tables 1 and 2 of the anonymous supplementary material, respectively. Setting a $1/5$ threshold is too strict (discarding tasks due to single model guess), while $5/5$ is too loose; **$3/5$ perfectly balances this trade-off**. Besides, human validation shows **100% precision** (all 20 filtered tasks confirmed as shortcut solvable) and **90.9% recall** (20/22 shortcut-solvable tasks correctly identified), indicating that the filter is conservative and reliable.
>
> > W4: the experiments do not include comparisons with other agent systems.
>
> Our primary evaluation intentionally fixes a simple shared scaffold so that differences more directly reflect the underlying model rather than architecture-specific optimizations. That said, we agree that alternative scaffolds are valuable and are integrating more agents into DSGym. In this rebuttal, we have incorporated the reviewer's suggestions and have integrated AIDE and obtained partial results:
>
> AIDE Results on DSPredict (public/private split):
> | Model                              | Max Time (Total/Exec) | Any Medal % | Above Median % | Avg Percentile | Valid Submissions % |
> | ---------------------------------- | --------------------- | ----------- | -------------- | -------------- | ------------------- |
> | Qwen3-235B-A22B-Instruct           | 5h / 2h               | 1.9 / 0.0   | 1.9 / 2.4      | 22.4 / 28.6    | 31.5 / 28.6         |
> | Qwen3-Coder-480B-A35B-Instruct | 5h / 2h               | 1.9 / 0.0   | 7.4 / 0.0      | 26.8 / 17.2    | 44.4 / 45.2         |
> | Kimi-K2-Thinking                   | 5h / 2h               | 1.9 / 0.0   | 3.7 / 0.0      | 33.1 / 20.6    | 13.0 / 9.5          |
>
> For additional context, we also evaluated Qwen3-Coder with AIDE on MLE Lite:
> | Model                              | Agent | Max Time | Any Medal % | Above Median % | Submissions % | Valid Submissions % |
> | ---------------------------------- | ----- | -------- | ----------- | -------------- | ------------- | ------------------- |
> | Qwen3-Coder | AIDE  | 5h / 2h  | 18.2        | 27.3           | 81.8          | 68.2                |
>
> Results show that integrating AIDE does not trivially solve the benchmark. (e.g., low medal rates and modest percentiles across models). Under our current setup, replacing the CodeAct scaffold with AIDE does not consistently improve outcomes, especially on DSPredict, suggesting that scaffold complexity alone does not explain the observed performance differences.

---

> > ### Author Rebuttal · Reviewer_U6JR · 2026-04-04
> >
> > Thank you for your response, and I will maintain my rating.

---

> > > ### Author Response · Authors · 2026-04-07
> > >
> > > We thank the reviewer for the follow-up and for considering our responses. In the rebuttal, we provided additional analyses and experiments addressing all raised concerns (e.g., SFT ablations, shortcut filtering sensitivity and validation, and integration of alternative agent systems). We also note that these clarifications were found helpful by other reviewers. If there are any remaining concerns that we have not fully addressed, we would greatly appreciate further clarification so we can improve the paper accordingly. We appreciate your time and feedback.

---

### Official Review · Reviewer_CRu6 · 2026-03-13

**Soundness:** 3
**Presentation:** 3
**Significance:** 2
**Originality:** 2
**Overall Recommendation:** 4
**Confidence:** 4

**Summary:**

This work presents DSGym, an extensible framework for data and execution management for scientific discovery tasks. DSGym unifies the format of a few representative datasets and supports large-scale parallel execution of generated programs in separated coding environments. With the proposed framework, the authors introduce DSGym-Tasks which incorporates representative datasets with stricter quality control as well as newly curated data for bioinformatics and tasks automatically sourced from Kaggle. The framework also supports synthetic data generation for fine-tuning open-weight LLMs. Experiments show that DSGym presents a challenging benchmark for frontier LLMs and the synthetic training data can effectively boost performance after fine-tuning.

**Compliance With Llm Reviewing Policy:**

Affirmed.

**Final Justification:**

Thanks for the response, I'll keep my positive rating.

**Key Questions For Authors:**

1. The stateful execution aspect of Section 2.3 seems trivial given DSGym is based on Jupyter kernel. Could you explain the significance/contribution of this feature?
2. Section 5.1 mentions that a temperature T=0 (Greedy Decoding) is used during evaluation. However, this is usually not the ideal inference setup for multi-turn agents in reasoning or coding tasks. Have you tried multiple evaluation runs at a higher temperature and report average performance?

**Limitations:**

yes

**Strengths And Weaknesses:**

Strengths:
1. The proposed framework and datasets are useful assets for the research community in benchmarking and data generation.
2. The evaluations settings are reasonable and authors provide valuable insight into benchmark construction, model performance, and failure modes.
3. The paper is well written and easy to follow.

Weaknesses:
1. Given previous file-based benchmarks such as ScienceAgentBench [1], the methodological contribution of this work is relatively limited.

Chen, Z., Chen, S., Ning, Y., Zhang, Q., Wang, B., Yu, B., ... & Sun, H. ScienceAgentBench: Toward Rigorous Assessment of Language Agents for Data-Driven Scientific Discovery. In The Thirteenth International Conference on Learning Representations.

---

> ### Author Rebuttal · Authors · 2026-03-30
>
> We sincerely thank the reviewer for the constructive review.
>
> > W1: Given previous file-based benchmarks such as ScienceAgentBench [1], the methodological contribution of this work is relatively limited.
>
>
> We respectfully believe the contribution is much broader than a file-based benchmark. **DSGym is a unified, extensible evaluation and training environment**. It introduces a unified task abstraction, a common agent/execution interface, and a reproducible gym-style environment that supports heterogeneous tasks under a single API. This standardization is intended to reduce fragmentation across prior benchmarks in task format, execution setup, and evaluation protocol, and to provide reusable infrastructure for both benchmarking and training.
>
> In addition, we view shortcut filtering as a methodological contribution to benchmark construction. Prior file-grounded tasks often implicitly assume that file access guarantees data dependence. In contrast, **DSGym explicitly shows that a substantial fraction of previous file-grounded tasks can be solved without data**, motivating a new benchmark construction step that removes frequently shortcut-solvable tasks.
>
> Finally, DSGym also expands the scope through new suites: expert-derived bioinformatics (DSBio) and real-world predictive modeling (DSPredict). Beyond evaluation, the same environment supports execution-grounded synthetic data generation for post-training, illustrating that DSGym serves as a reusable framework for both assessment and agent improvement.
>
> > L1: The stateful execution aspect of Section 2.3 seems trivial given DSGym is based on Jupyter kernel. Could you explain the significance/contribution of this feature?
>
> The core contribution of our execution layer is not the Jupyter kernel itself, but our robust, scalable manager-worker architecture that guarantees strict isolation, dynamic resource limits, and filesystem protection across parallel trajectories. This central manager spins up a fresh container per trajectory, maps read-only datasets and writable workspaces separately, enforces strict timeouts, and executes isolated artifact evaluation.
>
> > L2: Have you tried multiple evaluation runs at a higher temperature and report average performance?
>
> Thank you for this suggestion. Results on QRData show that evaluating 5 runs at $T=0.8$ yields stable performance that mirrors $T=0$, preserving model rankings. We will add this to the appendix.
>
>
> | Model | Temp = 0 | Temp = 0.8 |
> | -------- | -------- | -------- |
> | GPT-4o   |  60.2  |    59.8 ± 0.4   |
> | Qwen3-Coder  |  54.7  |  54.3  ± 0.5     |
> | Claude 4 sonnet  |  59.1  |   59.4  ± 0.5   |
> | GPT-5  |   61.7 |   61.4 ± 0.7   |

---

> > ### Author Rebuttal · Reviewer_CRu6 · 2026-04-03
> >
> > Thank you for the explanations. I still believe the methodological contribution of the proposed framework is relatively incremental, although it does bear engineering merit and practical value. My initial ratings has considered these contributions and thus I will maintain my score.

---

> > > ### Author Response · Authors · 2026-04-07
> > >
> > > We thank the reviewer for the thoughtful follow-up and for recognizing the engineering merit and practical value of our work! While we understand the concern regarding perceived incrementality, we respectfully believe that the combination of a standardized execution framework, shortcut-aware benchmark construction, and unified support for both evaluation and training represents a meaningful step toward more reliable and reproducible data science agent evaluation. We will further clarify these aspects and their distinctions from prior work in the final version. We appreciate the reviewer’s careful consideration!

---

### Official Review · Reviewer_7kvA · 2026-03-13

**Soundness:** 3
**Presentation:** 3
**Significance:** 3
**Originality:** 3
**Overall Recommendation:** 4
**Confidence:** 3

**Summary:**

This paper proposes DSGYM, a standardized, containerized framework for evaluating and training data science agents under stateful execution over real datasets. The framework unifies heterogeneous tasks under a common task abstraction and a CodeAct-like agent interface, uses isolated manager-worker containers with read-only dataset mounts, introduces shortcut filtering for tasks that can be solved without file access, and adds two new suites: DSBIO for domain-grounded bioinformatics analysis and DSPREDICT for Kaggle-style end-to-end prediction. The empirical study spans general analysis, scientific workflows, and modeling tasks; it highlights shortcut solvability, domain-grounding failures, and simplicity bias, and also demonstrates a 2,000-example execution-grounded SFT pipeline that improves a 4B model.

**Compliance With Llm Reviewing Policy:**

Affirmed.

**Final Justification:**

The author addressed my concerns.

**Key Questions For Authors:**

How sensitive is shortcut filtering to the specific model set and majority threshold? Have you tried smaller or different model pools, or compared the filter against human judgments on a subset?

**Strengths And Weaknesses:**

## Strengths

- Strong systems contribution. The manager–worker execution architecture, stateful kernels, read-only dataset mounts, and unified task abstraction are well aligned with realistic data-grounded evaluation, and they provide useful infrastructure beyond any single benchmark.

- Shortcut filtering addresses a real and underappreciated confound. The paper operationalizes data dependence by running five LLMs without file access and removing majority-solvable items; the accompanying experiments show that accuracy drops after filtering, supporting the claim that a non-trivial fraction of existing benchmark performance can come from memorization, priors, or prompt-level shortcuts rather than genuine data interaction.

- The benchmark coverage is meaningfully expanded. DSBIO brings domain-grounded scientific analysis into the evaluation loop, and DSPREDICT introduces end-to-end modeling tasks tied to real leaderboard metrics. I particularly appreciate that DSBIO is not just a collection of prompts: the appendix describes an explicit task-construction pipeline with publicly available datasets, expert-derived analyses, and iterative expert review.

- The paper goes beyond evaluation and demonstrates a plausible training use case. The execution-grounded synthesis pipeline, joint query-trajectory filtering, and 2,000-example DSGYM-SFT case study provide a concrete illustration of how the environment can also support agent training. The reported behavioral changes toward more iterative interaction are a nice addition.

## Weaknesses

- The shortcut filter is important but still heuristic. Because the filtering rule is a majority vote over five no-data model runs, it may still produce false positives or false negatives depending on the choice of models and threshold. The idea is strong, but I would like to see sensitivity analyses or partial validation against human judgment.

- DSPREDICT is more reproducible than an initial reading might suggest, but it still has a sustainability risk. The appendix already addresses reproducibility via a fixed offline leaderboard snapshot and a federated-download setup, which is good. However, the benchmark still depends on Kaggle infrastructure, private-leaderboard visibility, and competitions continuing to accept submissions; the paper explicitly notes that competitions may later be removed and replaced. This leaves some uncertainty about long-term stability.

- Some methodological details are present, but too buried in the appendix. This is especially true for DSBIO curation and failure analysis. Appendix E.2 describes the DSBIO pipeline in some detail, including the use of a primary expert, an independent expert, and a Gold Notebook solution. Appendix D.3 also explains that the failure study samples 50 failed trajectories per model/task family and uses two annotators with consensus resolution. These details materially improve my confidence in the work, and I think they should be surfaced more prominently in the main paper. The main remaining gap is the lack of inter-annotator agreement statistics.

- The SFT case study is promising but still limited. The current demonstration uses 2,000 examples and a judge-based filtering pipeline, but I did not see stronger ablations isolating the value of execution validation, the diversity filter, or the specific judge criteria, nor evidence of transfer across more backbones or onto DSPREDICT-style tasks.

- The comparisons are not perfectly apples-to-apples. Table 5 includes directly reported numbers for Jupyter Agent Qwen3 4B, and the appendix notes that Datamind-7B uses the checkpoint and system prompt from the original paper while most other models use the shared DSGYM prompt. This does not invalidate the results, but it makes some comparisons less clean than they first appear.

- Replicability details could still be expanded. The paper does report important runtime constraints, including per-turn limits, max turns, total DSPREDICT budgets, and A100 availability. Still, I would like clearer documentation of CPU/RAM limits, exact environment/image pinning, library versions, and dataset hashing/version identifiers.

- The current setup is strongly Python/Jupyter-centric. That is a reasonable starting point, but it may limit generality for communities where R or mixed-language workflows are common. The paper would benefit from a clearer discussion of what would be required to extend DSGYM beyond the current Python-first interface.

---

> ### Author Rebuttal · Authors · 2026-03-31
>
> We sincerely thank the reviewer for the constructive feedback. We provide sensitivity analysis, human validation, and SFT ablation experiments at: https://anonymous.4open.science/r/ICML2026_rebuttal_DSGym-2847/rebuttal_tables.pdf
>
> > W1: sensitivity analyses or partial human judgment for shortcut filtering.
>
> To validate shortcut filtering and the $\ge 3/5$ threshold, we provide Tables 1 and 2 in the supplementary materials for sensitivity analysis and human validation study. Setting a $1/5$ threshold is too strict (discarding tasks due to single model guess), while $5/5$ is too loose; **$3/5$ perfectly balances this trade-off**. Besides, human validation shows **100% precision** (all 20 filtered tasks confirmed as shortcut solvable) and **90.9% recall** (20/22 shortcut-solvable tasks correctly identified), indicating that the filter is conservative and reliable.
>
> > W2: depends on Kaggle infrastructure. some uncertainty about long-term stability.
>
> DSGym is designed as a live, evolving testbed with strict reproducibility guaranteed by versioned offline snapshots. While the benchmark relies on active Kaggle infrastructure, all reported numbers are anchored to a fixed offline snapshot, ensuring reproducibility regardless of future leaderboard drift. More broadly, the DSGym infra and the DSPredict construction pipeline make it straightforward to retire unavailable competitions and add newly available ones over time. As competitions close, we will routinely deprecate them and introduce new ones, releasing a newly versioned snapshot each time.
>
> > W3: Some details are buried in appendix. The main remaining gap is the lack of inter-annotator agreement
>
> We will elevate these methodological details to the main text. We also provide the inter-annotator agreement ratio and cohen's kappa for failure analysis. Two annotators independently labeled sampled failure trajectories, and disagreements were resolved through discussion for the final labels used. We report pre-consensus agreement metrics.
>
> - General Data Analysis (QRData + DAEval)
> | Model | Agreement. | κ |
> |---|---:|---:|
> | Claude4.5 Sonnet | 58.9% | 0.4312 |
> | Qwen3-coder-480B | 55.8% | 0.4097 |
> | Kimi K2 | 62.7% | 0.4821 |
> | GPT-5 | 60.9% | 0.4614 |
> - DSBio
> | Model | Agreement. | κ |
> |---|---:|---:|
> | Claude4.5 Sonnet | 60.6% | 0.4506 |
> | Qwen3-coder-480B | 57.4% | 0.4284 |
> | Kimi K2 | 64.9% | 0.4998 |
> | GPT-5 | 62.8% | 0.4796 |
>
> (Note: Agreement is higher on DSBio because failures are heavily dominated by domain-grounding errors)
>
> > W4: Stronger ablations, transfer across more backbones, DSPREDICT-style tasks.
>
> Thank you for the feedback! To strengthen this section, we provide additional ablation experiments (Table 3 of the anonymous supplementary materials):
> - **w/o execution validation/judge criteria/diversity filter**: Accuracy all drops on QRData and DABStep without each component. Removing execution causes the largest drop.
> - **Alternative backbone**: Applying DSGym-SFT-2k to Qwen2.5-7B yields similarly strong relative gains (35.04→55.38 on QRData).
> - **Eval on DSPredict** (Table 4 of the anonymous supplementary material): It's expected that SFT on analysis data does not improve DSPredict-Hard because those tasks are particularly hard and require qualitatively different (even domain-specific) modeling workflows. Importantly, DSGym directly supports synthesizing data for DSPredict-style modeling tasks, which we expect to yield stronger gains.
>
> > W5: Comparisons not perfectly apples-to-apples (Jupyter Agent / Datamind)
>
> We will explicitly clarify that these are informative external baselines, not perfectly controlled prompt-matched comparisons. Datamind requires its original system prompt to function (enforcing its trained output structure), and Jupyter Agent is strictly optimized for DABStep (tools, prompts, output structure, etc). This heterogeneity precisely highlights the need for DSGym—offering a unified substrate to evaluate divergent systems reliably.
>
> > W6: clearer documentation of CPU/RAM limits, environment/image pinning, etc.
>
> We'll add all the details to the appendix. Due to space limit, we only show partial details here: For DSPredict, we used a Dockerized testbed with 8 worker containers (CUDA 12.6, cuDNN 9) and 1 manager. Each worker was allocated 8 CPU cores and 24 GB RAM (cpus: 8.0, mem_limit: 24G). The container uses a non-root user and a fixed working directory (/app) to standardize execution behavior.
>
> > W7: what would be required to extend beyond the current Python interface.
>
> Extending DSGym beyond Python is straightforward due to its language-agnostic manager-worker architecture. The manager allocates containers based on specific type tags; adding an R environment simply requires (1) adding a new Docker worker image containing R and IRKernel, and (2) tagging the task metadata to route to this container.

---

> > ### Author Rebuttal · Reviewer_7kvA · 2026-04-03
> >
> > Thank the authors for the  detailed response, and it addressed my concerns to a certain extent. I will adjust my ratings accordingly.

---

> > > ### Author Response · Authors · 2026-04-07
> > >
> > > We sincerely thank the reviewer for the thoughtful review and for adjusting the rating! We appreciate the recognition, and will incorporate the suggested clarifications and additional analyses to further strengthen the paper in the final version.

---

### Decision · Program_Chairs · 2026-04-30

**Decision:**

Accept (regular)

**Comment:**

The paper presents a standardized framework for evaluating and training data science agents in self-contained execution environments. Authos contribute a set of data science tasks and enables agent training via execution-verified data synthesis pipeline. After reading the manuscript, all reviews, and the discussion, the AC thinks that the pros outweight the cons. Overall, the paper is beneficial for the agentic science domain and could be used to improve the discovery ability of foundation models.